

# Enhancing local action planning through quantitative flood risk analysis: a case study in Spain

Jesica T. Castillo-Rodríguez[1], Ignacio Escuder-Bueno[1,2], Sara Perales-Momparler[3], and Juan R. Porta-Sancho[4]

[1]Research Institute of Water and Environmental Engineering, Universitat Politècnica de València (UPV), Camino de Vera s/n, 46022 Valencia, Spain
[2]iPresas Risk Analysis, UPV Spin-off company, Avda. del Puerto 180 1B, 46023 Valencia, Spain
[3]Green Blue Management, Avda. del Puerto 180 1B, 46023 Valencia, Spain
[4]Oliva City Council, Pl. Ajuntament 1, 46780 Oliva, Spain

*Correspondence to*: J. T. Castillo-Rodríguez (jecasrod@upv.es)

**Abstract.** This article presents a method to incorporate and promote quantitative flood risk analysis to support local action planning against flooding. The proposed approach aims to provide a standardized framework for local flood risk analysis, combining hazard mapping with vulnerability data to quantify risk in terms of expected annual affected population, potential

injuries, number of fatalities, and economic damages. Flood risk is estimated combining GIS data of loads, system response and consequences and using event tree modeling for risk calculation. The study area is the city of Oliva, located in the Eastern coast of Spain. Results from risk modeling have been used to inform local action planning and to assess the benefits of structural and non-structural risk reduction measures. Results show the potential impact on risk reduction of flood defences, and improved warning communication schemes through local action planning: societal flood risk (in terms of

annual expected affected population) would be reduced up to 51% by combining both structural and non-structural measures. In addition, the effect of seasonal population variability is analyzed (annual expected affected population ranges from 82% to 107%, compared with the current situation, depending on occupancy rates in hotels and campsites). Results highlight the need for standardized methods for urban flood risk analysis replicability at regional, national and pan-European scale.

**Keywords.** River flooding, vulnerability analysis, risk analysis, flood risk management

## 1 Introduction

Floods are among the most damaging natural disasters in Europe and worldwide. In this paper, the need for improved and standardized quantitative flood risk analysis is identified, current and future challenges on flood risk reduction are acknowledged and a framework for flood risk analysis is presented and applied to a city as an example for enhanced local

flood risk management.



## 1.1 The need

In the period 1985-2015, Europe has suffered nearly 481 major flood events, with 3,136 fatalities, more than 12 million affected people and more than 123 US$ billion economic losses (Université Catholique de Louvain, 2015). The trend will probably continue to rise as floods and storms are expected to become more frequent and severe in Europe in the future

(UNISDR, 2009).

Urban areas concentrate population and economic activities thus presenting high flood vulnerability. Mediterranean cities are particularly affected by flooding as they are located next to rivers, in low-land areas and affected by flood events in ephemeral streams. In many Mediterranean cities, the combination of basin physical characteristics and intense and irregularly distributed rain generates frequent floods.

As an example, the Valencian region in Spain has suffered severe flood events in the last decades (Université Catholique de Louvain, 2015), highlighting the 1957 Turia river flood (with 77 fatalities) and the 1982 flood from failure of Tous dam (with 43 fatalities and more than 226,000 affected people).

As a result of the impact of past flood events and the need for reducing existent flood risk, the European Commission published the Directive 2007/60/EC on 6 November 2007 (European Parliament, 2007), aiming at reducing and managing

the risks that floods pose to human health, the environment, cultural heritage and economic activity. This Directive requires all Member States to assess risks related to water courses and coastlines, to develop hazard and risk maps and to apply measures to reduce flood risk.

This Directive was transposed into Spanish law by Royal Decree 903/2010, "Flood risk evaluation and management", which requires the definition of all areas with potential flood risk within the territory. This Decree establishes the content of hazard

and risk maps, along with flood risk management plans at river basin scale.

More particularly, in the Valencian region, the regional government developed PATRICOVA (Territorial Action Plan for Flood Risk Prevention) in 2003, a preventive tool with recommended actions for urban planning and flood risk reduction. Municipalities classified at medium and high flood risk levels are required to develop local action plans for flood risk management (in Spanish, 'Planes de Actuación Municipal ante el Riesgo de Inundaciones', herein denoted as PAMRI by its

acronym). PATRICOVA has been recently updated in 2015, incorporating new legislation and integrating recent advances in cartography.

Despite recent legislation and work conducted on flood risk management, there is still a need for local flood risk analyses to complete risk analyses developed at regional scale, to inform action planning and to better orientate risk reduction actions at urban scale. In most cases, developed flood risk analyses lack the required level of detail to support decision-making on local

flood risk reduction and planning.



## 1.2 The challenge

Flood risk management has acquired an important role since the European Floods Directive (Directive 2007/60/EC) and global strategies for flood risk reduction have evolved from focusing mainly on reducing the hazard (structural measures) to more holistic approaches including the combination of both hazard and impact mitigation.

'Think globally, act locally', the famous phrase attributed to René Dubos during the UN Conference on the Human Environment in 1972, emphasizes the importance of scale in dealing with environmental challenges. Unique physical, climatic, and cultural conditions appear at local scale and site-specific flood risk management is needed.

In the Valencian region, only 18 out of 136 local action plans for flood risk management have been developed and approved up to date. Despite the publication of some recommendations by civil protection on how to perform these plans, local

authorities do not have the information, know-how or experience on the required flood risk analyses to be developed.

The city of Oliva, located in the Eastern coast of Spain, belongs to the group of municipalities within medium to high flood risk levels. Located 70 km from Valencia, Oliva is affected by pluvial, river and coastal flooding and it is characterized by a complex and wide-ranging geography (e.g. hills up to 460 m.a.s.l., plains, coastal areas and wetlands). In addition, there is high seasonal variation in population (with 27,127 and 55,174 inhabitants of resident and seasonal population, respectively,

distributed across 60.1 km²).

After 28 years of the largest flood event in modern times in Oliva (accounting for the highest recorded rainfall rate at the Iberian Peninsula with 817 mm in 24 hours), local authorities face the challenge of mitigating flood risk through the development and implementation of a local action plan (as required by regional legislation), in line with other existent and ongoing structural measures for flood risk reduction.

Local authorities are aware of the need for improved flood risk management strategies and quantitative flood risk analysis arises as a helpful tool to support the definition of local flood risk management actions and strategies.

## 1.3 The opportunity

Flood risk is commonly expressed in terms of expected annual damage (in terms of potential affected population, number of fatalities or economic damage), combining data on flood hazard with information on exposure and susceptibility to that

hazard (Merz et al., 2010).

In this paper, we estimate flood risk and analyze how risk outcomes are affected by data and methods used for consequence estimations (e.g. population at risk, selection of depth-damage curves, etc.). A risk model capable of estimating annual risk for different scenarios is performed herein, based on results from hazard characterization and consequence estimations. A standardized framework for flood risk analysis is proposed.

This paper aims to contribute to find answers to the following questions: (i) how local flood risk management strategies may benefit from risk analysis; and (ii) which are the current barriers to standardized local flood risk analysis?



This paper analyses the city of Oliva (Spain) as an example. Although flood hazard mapping is available, a quantitative flood risk analysis had never been performed. The presented study is being used as a basis for developing a local action plan against flood risk. This 'science for policy' paradigm can be considered as a 'lighthouse' example for other cities that are required to develop their corresponding plans.

The study shows how hazard, exposure and vulnerability analyses provide valuable information for the development of a local action plan against flooding, for example by identifying areas with highest societal and economic risk levels.

## 2. Method

In this section, the applied framework (including tools and methods) for flood risk analysis is described. This framework for flood risk analysis is based on the method proposed by (Escuder-Bueno et al., 2012),  through the use of a risk model which

incorporates all information regarding loads, system response and flood consequences, and adapted to integrate GIS data into risk modelling.

Potential applications of this framework include local flood risk assessments such as those required by regional and national legislation in Spain after the 2007 European Floods Directive.

The steps of the proposed framework include:

— Phase I: Scope of the case study

   — Phase II: Review of available data

   — Phase III: Study of the system situation: definition of the Base Case

   — Phase IV: Flood events to be analyzed

   — Phase V: Risk model architecture

— Phase VI: Input data for the risk model

   — Phases VII and VIII: Risk calculation and representation

   — Phase IX: Risk evaluation

   — Phase X: Study of risk reduction measures

A GIS-based procedure to obtain input data for the risk model in Phase VI is proposed, aiming to provide a standardized

method for urban flood risk calculation and analysis.

### 2.1 Scope of the case study

The proposed framework aims at estimating flood risk in urban areas in terms of affected population, potential injuries, fatalities, and damage costs. It can be applied to analyze existent risk or to compare different scenarios to evaluate the impact of risk reduction measures.





## 2.2 Review of available data

Information on hydrologic studies, hydraulic modelling, flood defence response, population and land use data is required for characterizing loads, system response and estimating consequences from flooding.

GIS data on flood characteristics (e.g. flood depth, velocity, flooded area, etc.) and population and land uses is required to apply the procedure proposed in Phase VI.

## 2.3 Study of the system situation: definition of the Base Case

The Base Case may correspond with the scenario which represents the current situation of the system or a benchmark scenario for analyzing the impact of risk reduction measures.

## 2.4 Flood events to be analyzed

The range of plausible flood events should be considered, obtained from hydrologic studies, and analyzed through hydraulic simulations to characterize system response and flood extent.

Flood defence response should be incorporated when possible into hydraulic modelling to analyze the existent protection level and the impact on flood characteristics of their performance.

## 2.5 Risk model architecture

The use of risk models provides a logic and mathematically rigorous framework for compiling information of the system to estimate flood risk (Castillo-Rodriguez et al., 2014). The risk model is generally divided into three submodels: loads, system response and consequences.

The risk model can be represented by an influence diagram composed by nodes and connectors (Serrano-Lombillo et al., 2011). Nodes include information on loads (e.g. annualized probabilities of flood events), system response (e.g. failure probabilities of flood defence infrastructures) or consequences (e.g. results from consequence estimations in terms of affected population or economic damages).

In this paper, two generic schemes for defining the risk model architecture are proposed and shown in Fig.1. The first scheme can be used for analyzing flood risk for urban areas affected by river flooding from non-regulated systems. The second scheme should be used if potential failure of a flood defence (e.g. a dam) is incorporated into the analysis.

These two generic influence diagrams are an adapted version of those proposed by (Castillo-Rodriguez et al., 2014). These schemes allow to include the analysis of societal risk in terms of affected population, potential injuries, fatalities and economic damages.

For the first influence diagram, proposed to analyze flood risk in non-regulated systems, the following nodes are considered:



- Moment: this node includes information on probabilities for different time periods during the day (i.e. the probability of being during the day or at night). It can be used to later incorporate daily variability on potential consequences (e.g. affected population in industrial areas might change depending on the moment of the day).

- Season: this node includes information on probabilities for different seasonal periods during the year (i.e. the probability of being in summer or winter season). It can be used to later incorporate seasonal variability on potential consequences (e.g. affected population in urban areas might change if resident or potential population during summer is considered).

- Floods: this node includes information on probabilities for different flood events. A range of flood events is established, defined by minimum and maximum return periods. This range is divided into a number of intervals (e.g. 10, 20, etc.). Figure 2 shows how the range of plausible flood events is divided into intervals for risk calculations. These intervals are equally spaced in logarithmic scale along the given range of return periods. Each interval is represented by an annual exceedance probability (AEP), obtained by deducting AEP values of low and high interval limits. The example shows a range from 1-yr to 1000-yr flood events into 10 intervals. An additional interval is added to include flood events that exceed the 1,000-yr return period.

- Response: this node includes information on system response (e.g. peak flow river discharges).

- AP, NI, N, D: these nodes include information on consequence estimation in terms of affected population (AP), injuries (NI), fatalities (N) and economic damages in the urban area (D), respectively. Estimations for different flood events are obtained and incorporated into the risk model in each node.

For the second influence diagram, proposed to analyze flood risk in a regulated river system with a dam, the following nodes are considered:

- Moment, Season, Floods: these nodes are equivalent to the aforementioned described for the first influence diagram.

- Normal Operating Level (NOL): this node includes the water level at the reservoir in normal situation. For simplicity, it is assumed that this level is constant.

- Gate operability: this node includes probabilities for each possible combination of gate operability for dams with controlled outlet works.

- Routing: results from flood routing are included in this node for each flood event and gate operability combination. Two outcomes from flood routing analysis are required: the maximum water pool level at the reservoir and resulting peak flow discharge through outlet works for each combination.

- Response: for each load combination (represented by a maximum water pool level from flood routing), this node is used to consider two possible situations: failure and non-failure of the flood defence system, with complementary conditional probabilities of occurrence for each load combination. Hence, two branches emerge from this node to consider both options.



— Q failure: this node includes information on peak flow discharges resulting from flood defense failure. For non-failure cases, peak flow discharges from flood routing are used.

— AP_F, NI_F, N_F, D_F: these nodes include information on consequence estimation in terms of affected population (AP), injuries (NI), fatalities (N) and economic damages in the urban area (D) for flood events resulting from flood defense failure.

— AP_NoF, NI_NoF, N_NoF, D_NoF: these nodes include information on consequence estimation for flood events resulting from flood routing discharges for non-failure cases.

## 2.6 Input data for the risk model

Two software tools are proposed for input data processing and flood risk modelling:

— gvSIG Desktop (www.gvsig.com): an open source software, GNU / GPL license, with free use, distribution, study and improvement. Recently, gvSIG has been graduated as an OSGeo project (Open Source Geospatial Foundation). This GIS software tool was first developed by the regional government of the Valencian Autonomous Region (to be widely implemented in their regional and local systems) and now is further developed and promoted by the gvSIG Association.

— iPresas UrbanSimp (www.ipresas.com): a free software tool for flood risk calculation and analysis. This tool is a simplified version of iPresas Calc, first developed by the Polytechnic University of Valencia (UPV) and now by iPresas Risk Analysis (Spin-off UPV). iPresas Calc is a software tool that combines input data on flood hazard and impact to obtain expected annual risk (Serrano-Lombillo et al., 2011). Risk, in terms of expected annual societal or economic risk, is calculated by developing the event tree that considers all combinations of events that may lead to flooding.

The use of other available GIS tools can be applied within this framework (e.g. qGIS). In this paper, gvSIG has been proposed as it is being used by local governments in Spain.

## 2.6.1. GIS-based procedure for input data estimation and risk modelling

In this paper, a GIS-based procedure is proposed to integrate GIS data into the risk model. An overview of the proposed workflow is shown in Fig. 3.

This procedure shows the required steps to estimate flood consequences and to provide input data for the risk model in terms of affected population, potential injuries and fatalities and damage costs at local scale. This GIS-based procedure aims at boosting implementation of risk-informed local action plans through standardized consequence estimation and risk calculation.

The information required includes:



- Hydrological and hydraulic modeling. Flood characteristics should be estimated for each cell on the map representing the study area for different floods (a range of flood events with return periods up to, at least, 500- year is recommended). Two maps are required showing inundation depths and flow velocities for each cell.

- Consequence estimation. Several types of consequences per cell on the map are obtained. The impacts are then aggregated at municipality scale. The impacts include population exposed to flooding, injuries, potential fatalities and economic damages.

  o Affected population. Affected population should be obtained using census data (resident and seasonal population) and information on occupancy rates in hotels, campsites, etc.

  o Life-loss estimation. The life-loss estimation method proposed by MAGRAMA (Spanish Ministry of Agriculture, Food and Environment) for developing risk analysis at river basin scale is used. This method is based on the methodology proposed by DEFRA (Department for Environment, Food and Rural Affairs). Recent flood risk analyses have been conducted in Spain by applying this methodology, as for example in the Ebro River Basin (PREEMPT project "Policy-relevant assessment of socio-economic effects of droughts and floods"). For a detailed description on the method for estimating potential fatalities, the reader is referred to Wallingford et al (2006).

  o Economic damage estimation. It is based on the method used in PATRICOVA (Generalitat Valenciana, 2015). Potential direct economic damage costs are obtained using information on land use categories to define asset values and applying a depth-damage function, which estimates the expected damage for a given inundation depth.

- Risk modeling.

  o Input data on floods (exceedance probabilities), river discharge (system response) and estimated consequences (aggregated outcomes at municipality scale from GIS-data) is incorporated into the risk model to estimate societal and economic risk in terms of annual expected impacts.

**2.7 Risk calculation**

In this paper, flood risk is defined as the combination of the probability of a damaging flood event and potential consequences (Gouldby and Samuels, 2005; Schanze, 2006). Risk is estimated as the expected annual average damage of flooding in terms of societal or economic consequences. Hence, risk is obtained in terms of expected annual population affected (EAPA), number of injuries (EANI), fatalities (EAF) and damage costs (EAD).

The iPresas UrbanSimp software tool estimates risk by developing the event tree that includes all combinations of flood events, system response and related consequences.





## 2.8 Risk representation

Risk can be represented in F-N curves. The area under the curve is the annual expected number of fatalities, where the horizontal axis represents the level of consequences (e.g. number of fatalities, denoted as N) and the vertical axis represents the annual cumulative probability of exceedance (F) of each level of consequences.

Other type of consequences can be represented. These curves are then called F-D or F-AP, by representing economic damages (D) or affected population (AP), respectively.

## 2.9 Risk evaluation

Risk outcomes can be compared with tolerability recommendations if available. However, there still is a lack of tolerability criteria applied at local scale, although some recent examples can be found (Miller et al., 2015).

Tolerability recommendations for individual risk have been published by several authors and organisations (Vrijling, 2001). As an example, the United States Bureau of Reclamation suggests a limit of 0.01 fatalities per year for annualized societal risk (Hennig et al., 1997).

## 2.10 Study of risk reduction measures

Once risk is obtained for the Base Case, other scenarios can be analyzed to evaluate the impact of risk reduction measures.

New input data on loads, system response or consequences should be required and incorporated into the risk model. Risk outcomes for the new scenario are then compared with results for the Base Case.

## 3. Case study

An example of how the proposed framework can be applied is described in this section. The results have been used to guide the development of a local action plan.

### 3.1 Context

The municipality of Oliva is located in the eastern coast of Spain (Fig. 4), has about 27,127 inhabitants (distributed in several urbanized areas) and covers a total area of 60.1km².

The heaviest daily precipitations historically observed in Spain concentrate mainly on the coastal Mediterranean zone. Indeed, Oliva accounts for the most extreme daily precipitation record in the Iberian Peninsula with 817 mm on 3$^{rd}$

November 1987 (Ramis et al., 2013).

The mean annual precipitation reaches 850 mm. Flood events concentrate mainly during the rainy season from August to November. Table 1 shows a summary of most relevant flood events in Oliva.



The system is characterized by multiple river courses and brooks, with complex interconnections and a varying topography, including low-land areas and hills up to 460 m.a.s.l.

A dam is now under construction in Rambla Gallinera river course (a 62.5 m high concrete gravity dam, with a total reservoir capacity of 6.13 hm³ at dam crest level). This dam will provide flood protection up to a return period of 10 yr (Hijós Bitrián et al., 2010) and significant reduction on the peak flow discharges at this river course up to 56% (50-yr flood event). Discharges are also attenuated for floods with higher return periods, with a minimum reduction of 8.6% (5,000-yr flood).

Oliva is composed by several urbanized areas distributed within the municipality. The main area is located in the north-western part, concentrating 84.6% of resident population (59.6% of seasonal population). However, other areas located along the coast are relevant as population may increase by 23 times in some districts.

The selection of this study area is based on several reasons. First, the intensity and frequency of past flood events in the region are relevant. Second, good quality and up-to-date data are available on hazard, population and land use mapping. Additionally, the impact of structural and non-structural flood risk reduction measures has not been quantified so far. Finally, local authorities are currently involved in the process of developing the Municipal Action Plan against Flood Risk (denoted as PAMRI).

### 3.2 Scenarios

Four scenarios are considered for flood risk analysis as follows:

- Current situation (denoted as CS): this scenario represents the situation of the system before implementing structural measures for flood risk protection.
- Base Case (denoted as BC): this scenario represents the situation after dam construction. Differences in peak flow discharges in Rambla Gallinera are shown in Table 2 (e.g. from 282 to 182 m³/s for a 25-yr flood event).
- Implementation of a local action plan (denoted as CS+PAMRI): this scenario represents the situation after implementing a local action plan against flooding (PAMRI), which includes improved warning and communication schemes, public education campaigns and training of all actors involved in emergency management.
- Implementation of a local action plan after dam construction (denoted as BC+PAMRI): this scenario represents the situation after implementing both structural measures (dam construction) and the local action plan against flooding.

### 3.3 Data

Population and land use data are GIS-based. These data, provided by local government, is based on a yearly survey promoted by the regional government for all municipalities with less than 50,000 inhabitants (hereafter, EIEL database, by its acronym in Spanish). The municipality is distributed in 9,324 and 16,131 parcels of urban and rural land, respectively.





The EIEL database includes resident and seasonal population: "resident population" is obtained from census data and "seasonal population" is estimated from demographic trends observed in the last years during the summer season. It includes both resident and occasional population (but does not include hotel and campsite occupancy). For this analysis, the summer period ranges from mid-April to mid-September.

This database is completed with observations during site visits and other inputs from local authorities.

Two scenarios are considered for benchmarking: the Current Situation, denoted as CS, and the situation after dam construction or Base Case, denoted as BS.

### 3.3.1 Hazard estimation

The regional plan PATRICOVA defines 6 flood hazard levels (denoted from NP1 to NP6) based on probability of flood
occurrence (return periods of 25, 100 or 500 yr) and inundation depth (above/below 0.8 m). Flood hazard levels in Oliva were obtained in 2002 from an inundation study at regional scale and reviewed in 2013 (adding a new level to identify geomorphological hazards). However, resolution of GIS data used for the regional plan is too low (scale was 1:50,000 in 2002 and 1:25,000 for the updated version in 2013). In addition, the recent review did not consider new and ongoing structural actions for flood risk reduction.

In this paper, we used inundation data from a hydraulic model developed in 2010 by ACUAMED (Aguas de las Cuencas Mediterráneas S.A., public corporation and instrument of the Ministry of Agriculture, Food and Environment for Mediterranean River Basin Development Programme) and updated by TYPSA (consulting firm) in 2012 with a DEM (Digital Elevation Model) with a 5m horizontal resolution derived from LiDAR (Light Detection and Ranging o Laser Imaging Detection and Ranging) and corrected by site measures. However, only 3 flood events were modelled (return
periods of 25, 100 and 500 yr). Table 2 shows peak flow discharges for two scenarios: current situation and Base Case (with structural flood risk reduction measures including the dam under construction).

Inundation maps with results from a 2D hydraulic model in SOBEK (a modeling suite developed by Deltares), with runoff rates from HEC-HMS (developed by the U.S. Army Corps of Engineers, USACE) are used. These maps are raster-based, with a spatial resolution of 20 m×20 m. Data on flood depth and velocity are available at each grid-cell for the three return
periods. Inundation map for the 500-yr flood event and hazard level map as defined by PATRICOVA are included in supplementary material for the current situation. Hazard level map for the current situation is also shown in Fig.5.

Results from hazard analysis show that 10% of resident population is located in low frequency flood areas, against a 15% of resident population located in high frequency areas (25-yr flood event). Around 14,000 are located in NP1 areas, "high frequency-high flood depth" category (flood depth greater than 0.8 m for the 25-yr flood event).

### 3.3.2 Consequence estimation

The municipality is divided into urban and rural parcel sub-areas and information from EIEL database is available in GIS format.





The number of resident and potential (seasonal) inhabitants in each parcel is obtained by multiplying the number of registered households and the corresponding density value (inhabitants/household). In addition, population in camping areas and hotels is considered based on the maximum capacity and hotel occupancy rates in the Valencian region (2013 Database from National Statistics Institute). These rates are assumed to be, in average, 35% and 75% in winter and summer seasons, respectively.

Table 3 summarizes the results of affected population. A 500-year flood has a 0.2 % probability of occurring in any given year, and could cause roughly 22,890 affected population during summer season, if there is no flood protection provided by structural measures for flood risk protection (Current Situation).

Estimation of potential life-loss is based on the method proposed by DEFRA (Wallingford et al., 2006), including the following assumptions:

— An average debris factor (DF) equal to 0.5 is used to estimate hazard rates.

— A vulnerability area factor (AV) equal to 6, 7 and 8 is used for multi-storey buildings, residential areas and campsites, respectively.

— A population vulnerability factor (Y) of 0.2 is used based on census data (i.e. percentage of population aged 65 years and over).

We calculated the potential direct economic damage using information on land use classes to define asset values and a generic depth-damage function, which estimates the expected damage for a given inundation depth.

It is essential to adjust asset values to the regional economic situation and property characteristics (Jongman et al., 2012). Therefore, asset values and a generic stage-damage function used in regional studies for flood risk planning are considered in this case study. A sensitivity analysis has been included to analyze the impact of the selected stage damage function and reference asset values. Different stage damage functions would impact on consequence estimation results as later described in Sect.3.7.

Other direct costs such as destruction of vehicles, damage to infrastructure, livestock or business interruption are not considered. Indirect costs are considered based on factors used by regional planning, set as 7% of direct costs for the city of Oliva (it includes aspects such as population, employment and number of households within the urban area).

Table 3 summarizes the results of consequence estimation. A 500-year flood could cause roughly 9 potential fatalities and 52M€, if there is no flood protection provided by the dam.

The impact of implementing a local action plan against flooding (PAMRI) is analyzed by estimating risk for both CS and BC scenarios, with the following changes on consequence estimation from improved warning systems and communication schemes:

— A lower rate of vulnerability area factor (AV) is considered. Hence, values change to AV=5 in urbanized areas with multi-storey buildings, AV=6 in residential areas and AV=7 in campsites.

— A reduction on economic damages is assumed based on damage avoided when a warning lead time of, at least, 2 hours is provided. For a 80% rate of warning coverage (proportion of covered properties), 100% rate of service





effectiveness (proportion of flooded serviced properties that were sent a timely, accurate and reliable flood warning), 80% rate of availability (proportion of flooded services properties that received warning), 85% rate for ability (proportion of residents able to understand and respond to such a warning), and 85% rate for effective action (proportion willing to take effective action or which have actually taken effective action), a percentage of damage

5        reduction of 18% is assumed for flood depths below 1.2 m (Parker et al., 2005).

## 3.4 Model

For this case study, the risk model architecture shown in Fig.6 is used. Dam failure flood events are not modelled. Flood characteristics after dam construction (Base Case) are considered based on flood routing.

In order to determine societal or economic risk, the choice of a wide range of flood events is important. However,

availability covers from 25 to 500 yr. Vulnerability was estimated for 25-, 100-, and 500-year flood events, based on the proposed framework in Sect.2.

Given the discrete set of flood events, the range of plausible flood events is divided into 20 intervals, obtaining expected damage for each interval by interpolating input data for the three available flood events.

The impact of a 1-year-flood event is assumed to be zero for the current situation. A flood protection level of 10-yr is

considered for the Base Case.

## 3.5 Results

Results are summarized in this section. Table 5 shows results in terms of expected annual population affected (AEAP), number of injuries (AENI), fatalities (AEF) and damage (AED).

Risk outcomes for the current situation show societal risk levels up to 2,370 of annual expected affected population and 0.56

fatalities per year. Considerable risk reduction can be achieved by implementing planned structural measures and AEAP would be reduced in 1,202 inhabitants per year.

In addition, results reflect the combined effect of both structural measures and improved warning and evacuation systems for flood risk reduction. Societal risk after dam construction and implementation of the local action plan might change from 0.56 to 0.24 fatalities per year. Economic risk in terms of annual expected damages would vary from 6.11 to 1.89 M€ per year.

It is noted that at this stage, only direct benefits (such as the reduction in flood damage and improved warning systems) are included in the analysis of the impact of implementing a local action plan. Other benefits such as improved risk awareness or reduction on economic damages to vehicles and local businesses could be considered in future analyses.

Figure 7 shows F-AP, F-N and F-D curves for all scenarios.

## 3.6 Risk mapping

Different hazard and risk maps have been developed for the city of Oliva to support local action planning against flood risk. Recommendations published by the RISKMAP project (www.risk-map.org) have been considered for elaborating these



maps. An example is provided as supplementary material to this paper (affected population for the 500-yr flood event for the current situation).

## 3.7 Sensitivity analysis

The effect on societal and economic risk of several factors has been assessed in this study. Input data for the risk model has been modified and risk estimations obtained for each case.

### 3.7.1 Effect of selected flood protection level

In general, a flood protection level represents how well protected any given area is against flood damage. For example, a 10-yr flood protection system protects an area against anything equal to or smaller than a 10-yr flood.

Risk analysis for the current situation has been performed by assuming that flood damage is zero for a 1-yr flood event. In this section, the effect of selected vs. actual flood protection level is analyzed.

Results from Aqueduct Global Analyzer Database at regional scale are available for different protection levels. Model setup, results and limitations of available estimations in this database can be found in (Ward et al., 2013b; Winsemius et al., 2013). Table 5 shows the results for the Valencian region from this database, accounting that there is a region-wide average protection level of 2-, 5-, and 10-yr, respectively.

Risk estimations for the current situation have been obtained for three different protection levels (i.e. assuming that flood damage is zero for 2-, 5-, and 10-yr flood events) and are summarized in Table 5.

Results show that societal risk in terms of AEAP would change from 2,370 to 1,557 inhabitants/yr if a 10-yr protection level is assumed.

Since there is no information on system response for flood events with low return periods (hydraulic modeling was conducted from 25 up to 500-yr flood events), it is noted that risk estimated for the current situation might be overestimated for this case study. Further research on system response for high-frequency flood events would be of paramount interest.

We highlight that societal risk for the city of Oliva represents a significant percentage of total flood risk at regional scale. Despite it accounts for 1% of resident population at regional level, societal risk ranges from 5% to 10%, depending on the protection level, as shown in Table 5.

Results from this study can be used to validate/update available information in global databases.

### 3.7.2 Effect of including seasonal population variability on societal risk

The impact of occupancy rates in hotels and campsites on societal risk has been assessed. Two situations are considered:

- Occupation rates set to zero. Only census data and people in dispersed housing are used for estimating population at risk.
- Occupation rates set to maximum plausible values (50% in winter and 100% in summer).





Incorporating the above input data on consequence estimation into the risk model, societal risk results for these two scenarios show that values would range from 1,940 affected population/yr and 0.38 lives/yr (low occupancy) to 2,529 affected population/yr and 0.63 lives/yr (high occupancy). Results show that affected population increases in 450 inhabitants/yr when comparing zero occupancy's and the current situation's results.

These results show the importance of analyzing not only census data but considering potential population in hotels and campsites. This population group is of high relevance in touristic cities, as it is the case in the Mediterranean coast of Spain.

### 3.7.3 Effect of population trends on societal risk

Flood risk in the future can be influenced by either climate change, which may increase or decrease the frequency and severity of flooding; or by socio-economic changes, such as ageing population (or decline) and economic growth.

In this section, socio-economic change is considered. The database of Shared Socioeconomic Pathways (SSPs) developed by IIASA (International Institute for Applied Systems Analysis) is used for defining population trends in Oliva, based on national population trends for Spain in 2030 and 2050 (Nakicenovic et al., 2013). This database has been also used in recent local flood risk assessments (Ward et al., 2013a) in Europe.

Resident and seasonal population is increased by a factor of 1.06 and 1.13 in 2030 and 2050, respectively. Estimating 15  societal risk for these two scenarios, risk would range from 2,370 affected population/yr and 0.56 lives/yr (current situation) to 2,616 affected population/yr and 0.61 lives/yr in 2050.

Results show that attention should be paid on future population trends and urban developments to update vulnerability assessments.

### 3.7.4 Effect of selection of depth-damage curve on economic risk

The stage damage function used for this case study is the curve proposed in PATRICOVA (Generalitat Valenciana, 2015) for meso-scale flood risk analysis in the Valencian region, denoted as CS-curve. This curve has been compared to other relative (in percentage of damage) depth-damage functions. These curves are shown in Fig.8 and include:

- MAGRAMA: Stage damage function proposed by MAGRAMA for flood risk analysis and mapping at river basin scale (MAGRAMA, 2013).
- EGM: Stage damage function proposed by USACE, based on empirical data from flood events from 1995 to 1997, developed for nation-wide applicability in flood damage reduction studies (USACE, 2000).
- HYDROTEC: Simple curve used for some flood action plans in Germany (Merz and Thieken, 2009).

These generalized functions represent some of the existent depth damage curves for assessing urban flood damage. It is noted that CS and MAGRAMA curves may tend to overestimate costs. However, both curves include content damage in 30  reference costs to be multiplied by damage percentages thus no additional costs to content should be considered. Results from applying the MAGRAMA curve to the current situation are summarized in Sect. 3.7.5.



Other factors influence flood damage such as flow velocity, contamination, building materials and quality, etc., but are not considered in this analysis.

Depth-damage functions should ideally be developed for specific characteristics of local building types. Some examples of site-specific stage damage functions in Spain can be found (Velasco et al., 2015). However, the development of synthetic

curves for each urban area requires an exhaustive field work, data gathering and later analysis, not feasible in many cases.

Defining regional specific stage damage functions for most relevant land use types would be desirable and useful for comparison among cities. In addition, detailed local data on building types (not available for this study) would be of interest to estimate direct flood damages in future analyses.

### 3.7.5 Effect of asset values on economic risk

Reference values per land use type used for this case study correspond with rates proposed in PATRICOVA (Generalitat Valenciana, 2015). Direct costs for cleanup expenses, emergency prevention actions, and other related costs are not included. Table 6 shows reference values per land use type proposed by MAGRAMA for river basin flood risk analysis and mapping. These rates include replacement costs for infrastructure, content and vehicles. Therefore, reference values differ from those proposed in PATRICOVA (Generalitat Valenciana, 2015).

By matching land use categories defined by both sources (Generalitat Valenciana, 2015; MAGRAMA, 2013), risk is estimated for the current situation by adopting new reference costs and the stage damage function shown in Fig. 8.

It is noted that economic risk outcomes are highly sensitive to the stage damage function and reference values adopted, since economic risk would increase from 6.11 M€/yr (for the current situation) to 180.4 M€/yr (for the current situation, but using proposed values by MAGRAMA). These results show the need for standardized stage damage functions and reference asset

values at regional and national scale.

Both local and river basin flood risk analysis should consider the same method for economic consequence estimation to allow comparative analysis, to upgrade current and future flood risk plans and to develop cost-benefit analysis for prioritizing flood risk reduction measures.

### 4. Discussion

In this section, limitations of the proposed framework and implications of flood risk analysis outcomes to local action planning are described, along with recommendations towards standardized flood risk analysis.

### 4.1 Limitations

The analysis framework used in this study is relatively straightforward, but it does allow to analyze risk and to assess the impact of different scenarios. It is proposed as a standardized framework for enhancing local flood risk analysis at regional,

national and pan-European scale.



Results from the case study demonstrate its applicability and usefulness to support decision making for local action planning. However, the following remarks are made:

— Type of flooding. In this paper, we analyze river flooding. Integrating multiple hazards would be of high interest (e.g. to analyze the influence of sea water levels in boundary conditions).

— Flood hazard. It is recognized that over-estimations of annual risk between 33% and 100% have been reported in other studies when only three return periods are used (Ward et al., 2011). Therefore, results suggest that results for the case study could benefit from paying more attention to the potential damage caused by high-probability flood events.

— Consequence estimation. A generic relative stage damage function is used for the case study, based on methods used for regional planning.

— Uncertainty. Sources of natural and epistemic uncertainty include the occurrence of flood peak and flood volume, hydrograph shape, time and spatial rainfall distribution, lack of data on detailed building typology, among other factors. Sensitivity analyses indicate that societal risk is dominated by seasonal variability on population and potential population in high vulnerable areas.

## 4.2 Local action planning implications

The application of the proposed framework for quantifying local flood risk for the city of Oliva represents a novel analysis at regional and national scale.

The following recommendations were made to local authorities for defining strategies for local action planning, derived from outcomes of conducted flood risk analysis:

— Definition of specific public education campaigns for resident and seasonal population is needed, with emphasis in high vulnerable groups (e.g. the elderly, schools and campsites).

— A procedure to formally reporting flood events, damages and effect of communication and evacuations procedures is required for future updates of hazard and vulnerability analysis.

— Verification of established communication schemes between regional and local authorities, and with emergency and civil protection services is needed to ensure effectiveness of non-structural measures for flood risk reduction.

— Potential locations for assembly points and helicopter landing sites have been set based on population clusters, hazard maps, and available evacuation routes. These sites should be verified and reviewed in future updates.

— Data gathering on additional urban characteristics (e.g. building typology, daily variability of population in industrial and commercial areas, etc.) is encouraged to upgrade risk analyses and provide improved outcomes for decision making.



— Future flood risk mitigation measures should be planned to reduce annual expected affected population. If societal risk tolerability criteria are not available or applicable, strategies might be defined in accordance with the ALARP principle (As Low As Reasonably Practicable), used in safety applications (SPANCOLD, 2012).

The proposed framework for flood risk analysis will allow updating in future reviews of the local action plan and assessing

the impact of future risk reduction actions at local scale.

## 4.3 Recommendations for flood risk analysis and management

Based on results from this analysis, we recommend that quantitative risk analyses become the basis for developing local flood risk management plans. Specific recommendations include:

— Upgrading hydraulic modeling to a broad set of flood events for hazard mapping, and analyzing not only river
flooding but also pluvial or coastal flooding.

— Improved land use data gathering at local scale for better analyze life-loss and economic consequences from flooding.

— Improved data gathering on population characteristics and distribution at local scale.

— Standardized relative stage damage functions and reference costs at national scale.

## 15 5. Conclusions and the way forward

Quantification of societal and economic risk is not required by current legislation and is relatively novel in local flood risk management. Therefore, there is a lack of standardized methods or tools for local flood risk analysis.

The main scope of this study was to propose a common framework for quantitative flood risk analysis at local scale and to analyze urban flood risk for the city of Oliva.

Local authorities are currently developing the local action plan against flooding for Oliva, as required by regional legislation. Results from the flood risk analysis described in this paper have informed local authorities to define strategies and to make decisions on upcoming public education campaigns and training activities. In addition, assembly and monitoring points have been identified based on conducted flood risk analyses and identified hazard levels.

Results show that societal and economic risks, while considerably reduced from planned structural measures (a dam is now

under construction), are still significant, but they can be further reduced through local action planning.

The results of this study show that improved communications schemes and verified warning systems could significantly decrease flood risk. These results can be used to support risk communication and increase risk awareness.

Sensitivity of existent flood risk to vulnerability estimations has been addressed and future scenarios have been compared with the current situation.



Existent hazard maps have been used for identifying affected areas. A broad range of hydraulic simulations, covering 5 to 7 return periods would be desirable (Ward et al., 2011). In addition, further research to analyse dam failure scenarios and the impact of climate change on system response is recommended.

A more comprehensive risk analysis can be carried out to include other sources of flood hazards such as pluvial or coastal
flooding. The combination of multiple flood hazards should be taken into account in future risk analyses.

Further research on the impact on risk of mitigation measures (including data gathering through workshops or surveys) could inform local actors on the definition of incentives for flood risk mitigation.

The presented approach can be potentially applied by other cities to perform similar flood risk analysis. There is still a long way to go in the development and implementation of local action plans against flooding. The study described in this paper
aims to become a reference example for other cities towards improved flood risk management.

**Team list**

Ms. Jesica T. CASTILLO-RODRÍGUEZ. Civil Engineer (EQF7, 2008) and MEng Hydraulic Engineering and Environment (EQF7, 2012); researcher and PhD student at the Universitat Politècnica de València (UPV), with more than five years of research experience in flood risk analysis and risk-informed dam safety management, through several national and EU-
funded research projects and private contracts, as well as developing local action plans against flood risk. She is currently developing her PhD on the application of risk analysis techniques toward an integrated flood risk management by incorporating multiple hazards (natural and manmade risks).

Prof. Dr. Ignacio ESCUDER-BUENO. MSCE in Civil Engineering (1996) and PhD (2001); Professor at the Universitat Politècnica de València (UPV), and promoter and founder associate of iPresas (a technology based SPIN-OFF company of
the UPV); Secretary-General of International Commission on Large Dams (ICOLD) European Club since 2010 and Full Elected Member of the Spanish National Committee on Large Dams (SPANCOLD) since 2007; Visiting Professor at University of Maryland (2014) and Utah State University (2006) and Teaching Assistant at University of Wisconsin (1995-1996); coordinator of national and EU-funded research projects on risk analysis, critical infrastructure management and energy efficiency (e.g.E²STORMED).

Dr. Sara PERALES-MOMPARLER. Civil Engineer (EQF7, 2002) and PhD (2015) from the Polytechnic University of Valencia (UPV); Independent consultant (Green Blue Management); Director of PMEnginyeria, consultancy firm specialized in Sustainable Drainage Systems (SuDS), from 2006 to 2013; she worked for Sinclair Knight Merz (SKM), an international consultancy firm, both in the United Kingdom and New Zealand from 2004 to 2006; she has participated in several EU-funded research projects on flood risk analysis, SuDS, water and energy efficiency (e.g. AQUAVAL,
E²STORMED).

Mr. Juan Ramón PORTA SANCHO. Bachelor's Degree in Technical Architecture (EQF6, 1983 and 2011) and Master's Degree in Urban Emergency Prevention and Management (1992) from the Polytechnic University of Valencia (UPV);



Technical Architect at the Oliva City Council since 1992; Head of Urban Management and Civil Protection Division; he has coordinated and supervised urban planning studies and works for nearly 30 years (co-author of the Territorial Emergency Plan of Oliva, 1994), and, more recently, projects for implementing GIS-based systems at local scale and flood risk studies in Oliva; he is currently coordinating the development of the Municipal Action Plan against Flood Risk for the city of Oliva,

as Technical Director. He has also participated in several EU-funded research projects (e.g. QUATER, DAMAGE).

## Code availability

iPresas UrbanSimp is available for download at www.ipresas.com

## Data availability

Flood hazard maps and the local action plan are available to the public at www.oliva.es [In Spanish]. GIS-based local data is

not publicly accessible due to its protection level (owned by local authorities).

## Author contribution

J.T. Castillo-Rodríguez and I. Escuder-Bueno proposed the method and tools for flood risk analysis. J.T. Castillo-Rodríguez performed analyses and prepared the manuscript, in close collaboration with all co-authors (discussion of methods and tools, results and conclusions). S. Perales-Momparler has developed the Local Action Plan (PAMRI) for the city of Oliva including

outcomes from this study, following a risk-informed approach, with contributions of all co-authors in a joint science-policy effort. J.R. Porta-Sancho has contributed to the acquisition, analysis of data, and interpretation of results, and supervised all activities concerning the development and approval of the Local Action Plan (PAMRI).

## Acknowledgements

This research was conducted within the framework of the INICIA project, funded by the Spanish Ministry of Economy and

Competitiveness (BIA2013-48157-C2-1-R). The article processing charges for this open-access publication will be covered by the INICIA project. We would like to thank the City of Oliva for their willingness to share data, knowledge and experience with the authors, and for initiating this risk-informed journey.

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



| Date (yyyy-mm-dd) | 1987-11-03 | 1997-12-04 | 1972-11-29 | 1997-06-18 | 2002-03-30 | 1996-09-11 | 2002-05-05 |
|---|---|---|---|---|---|---|---|
| Precipitation in 24 h (mm) | 817 | 378 | 354 | 288 | 220 | 197 | 188 |

**Table 1. Summary of recent (most relevant) flood events for the case study.**





| River course | Current situation (Scenario 0): CS | | | Base Case (Scenario 1): BC | | |
|---|---|---|---|---|---|---|
| | Return period (yr) | | | | | |
| | 25 | 100 | 500 | 25 | 100 | 500 |
| Piles | 84 | 153 | 247 | 84 | 153 | 247 |
| Fonts | 54 | 107 | 186 | 54 | 107 | 186 |
| Algepsar | 7 | 11 | 23 | 7 | 11 | 23 |
| Frares | 4 | 7 | 16 | 4 | 7 | 16 |
| Alfadalí | 21 | 34 | 82 | 21 | 34 | 82 |
| Cementeri | 2 | 4 | 8 | 2 | 4 | 8 |
| Gallinera | 282 | 462 | 1025 | 182 | 284 | 829 |
| Benirrama | 16 | 28 | 63 | 16 | 28 | 63 |
| Bullent | 102 | 173 | 399 | 102 | 173 | 399 |
| Molinell | 84 | 146 | 318 | 84 | 146 | 318 |

**Table 2. Simulated peak flow discharges per river course (SOBEK model) [m³/s].**





| Season | Return period (yr) | Current situation (Scenario 0): CS | | | | Base Case (Scenario 1): BC | | | |
|---|---|---|---|---|---|---|---|---|---|
| | | AP | NI | N | D (M€) | AP | NI | N | D (M€) |
| Summer (seasonal population) | 25 | 7795 | 85 | 2 | 10.86 | 5596 | 59 | 1 | 5.27 |
| | 100 | 13269 | 158 | 3 | 22.20 | 9850 | 109 | 2 | 12.15 |
| | 500 | 22890 | 341 | 9 | 52.03 | 18754 | 270 | 7 | 42.39 |
| Winter (resident population) | 25 | 1873 | 25 | 1 | 10.86 | 1572 | 22 | 0 | 5.27 |
| | 100 | 3428 | 51 | 1 | 22.20 | 2539 | 35 | 1 | 12.15 |
| | 500 | 6282 | 110 | 3 | 52.03 | 4497 | 80 | 2 | 42.39 |
| | | CS+PAMRI | | | | BC+PAMRI | | | |
| | | AP | NI | N | D (M€) | AP | NI | N | D (M€) |
| Summer (seasonal population) | 25 | 7795 | 73 | 1 | 9.91 | 5596 | 51 | 1 | 4.73 |
| | 100 | 13269 | 136 | 3 | 20.26 | 9850 | 94 | 2 | 10.95 |
| | 500 | 22890 | 293 | 8 | 47.61 | 18754 | 232 | 6 | 38.92 |
| Winter (resident population) | 25 | 1873 | 22 | 0 | 9.91 | 1572 | 19 | 0 | 4.73 |
| | 100 | 3428 | 44 | 1 | 20.26 | 2539 | 30 | 1 | 10.95 |
| | 500 | 6282 | 95 | 3 | 47.61 | 4497 | 69 | 2 | 38.92 |

Note: AP=Affected population; NI=number of injured people; N=fatalities; D=damage costs in M EUR.

**Table 3. Estimated impact per scenarios CS and BC for the three flood events.**





| | Scenario 0: CS | Scenario 0 + PAMRI | Scenario 1: BC | Scenario 1 + PAMRI |
|---|---|---|---|---|
| Societal risk (AEAP) [inhabitants/yr] | 2370 | 2370 | 1168 | 1168 |
| Societal risk (AENI) [injured inh./yr] | 28 | 24 | 21 | 18 |
| Societal risk (AEN) [fatalities/yr] | 0,56 | 0,48 | 0,28 | 0,24 |
| Economic risk (AED) [Million EUR/yr] | 6,11 | 5,57 | 2,10 | 1,89 |

Note: CS=current situation; BC=Base Case; PAMRI=Local Action Plan; AE=annual expected; AP=Affected population; NI=number of injured people; N=fatalities; D=damage costs.

**Table 4. Results from risk model per scenarios CS, BC and effect of local action plan.**



| | Oliva (Scenario 0: CS) | | Valencia (region) | | Comparison Local/Region | |
|---|---|---|---|---|---|---|
| Flood protection level (yr) | Societal risk (AEAP) [inhabitants/yr] | Economic risk (AED) [MEUR/yr] | Societal risk (AEAP) [inhabitants/yr] | Economic risk (AED) [MEUR/yr] | %AEAP | %AED |
| 1 | 2370 | 6,11 | No data | No data | - | - |
| 2 | 2279 | 5,88 | 47600 | 746.24 | 4.8% | 0.8% |
| 5 | 1991 | 5,16 | 29000 | 537.94 | 6.7% | 0.9% |
| 10 | 1557 | 4,07 | 15800 | 348.04 | 9.8% | 1.2% |

Note: CS=current situation; AEAP=annual expected affected population; AED=annual expected damage costs.
**Table 5. Effect of the selection of flood protection level.**





| Land use type | Storage | Commercial | Cultural | Industrial | Office | Households | Sanitary | Agricultural |
|---|---|---|---|---|---|---|---|---|
| GVA | 11.25 | 34.55 | 34.55 | 11.25 | 34.55 | 68.7 | 34.55 | 0.8 |
| MAGRAMA | 150 | 380 | 200 | 450 | 380 | 350 | 200 | 5 |

**Table 6. Reference costs in EUR/m² in urban areas: GVA (2015) and MAGRAMA (2013).**



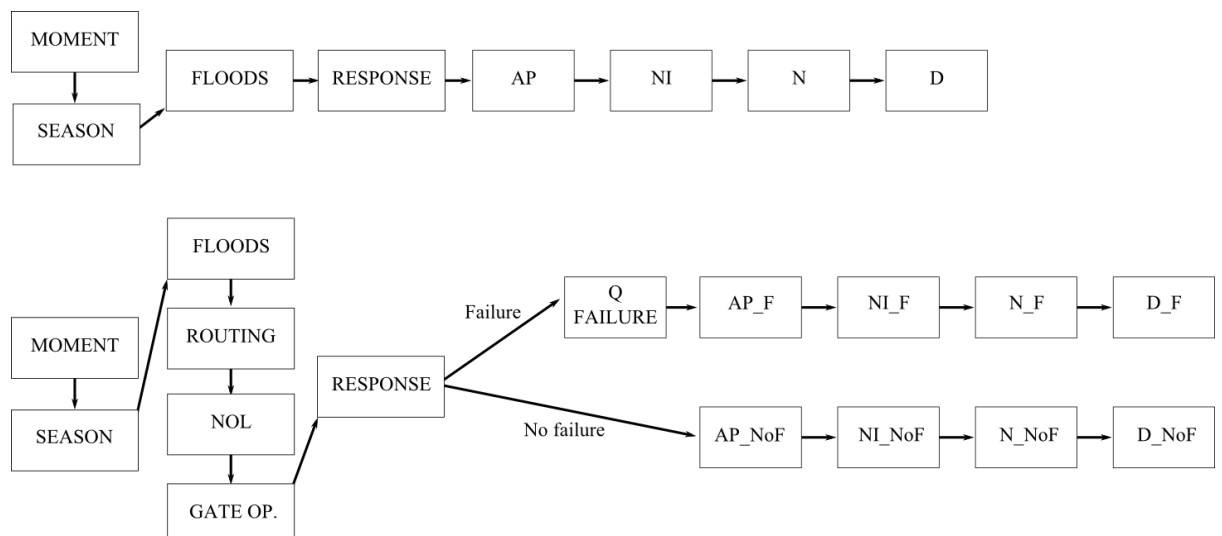

**Figure 1: Generic risk model architecture: non-regulated river system (a) and regulated river system (b).**



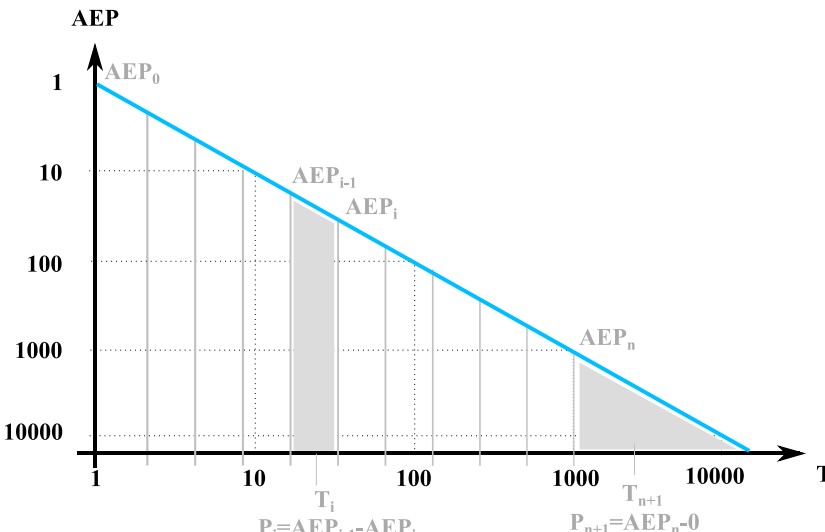

**Figure 2: Generic division of the analyzed range of flood events.**







**Figure 3: Flowchart of data and models.**



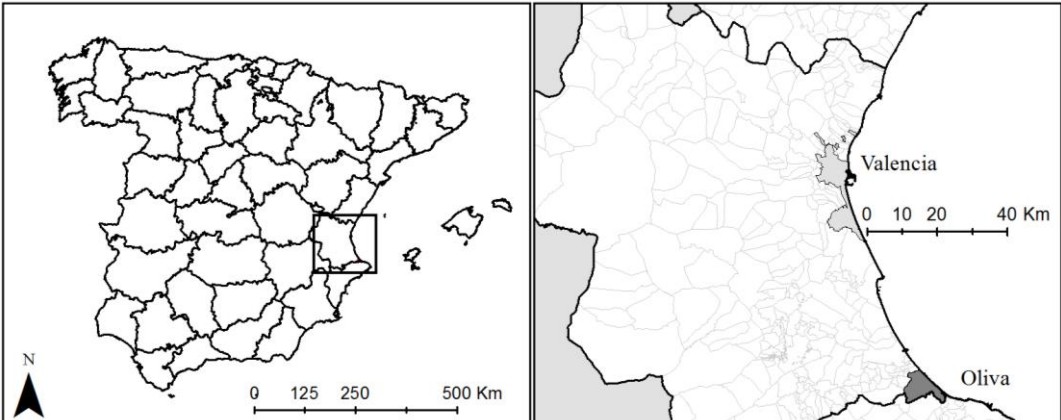

**Figure 4: Location of the case study area at national (left) and regional (right) scale.**



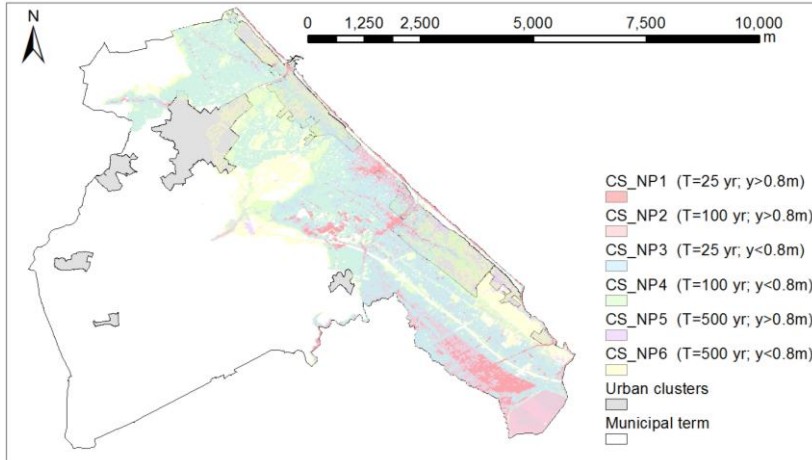

**Figure 5: Hazard level map for Scenario 0 (Current situation).**



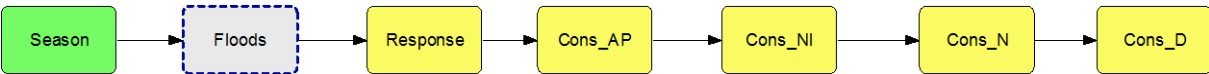

**Figure 6: Risk model architecture for the case study of Oliva.**





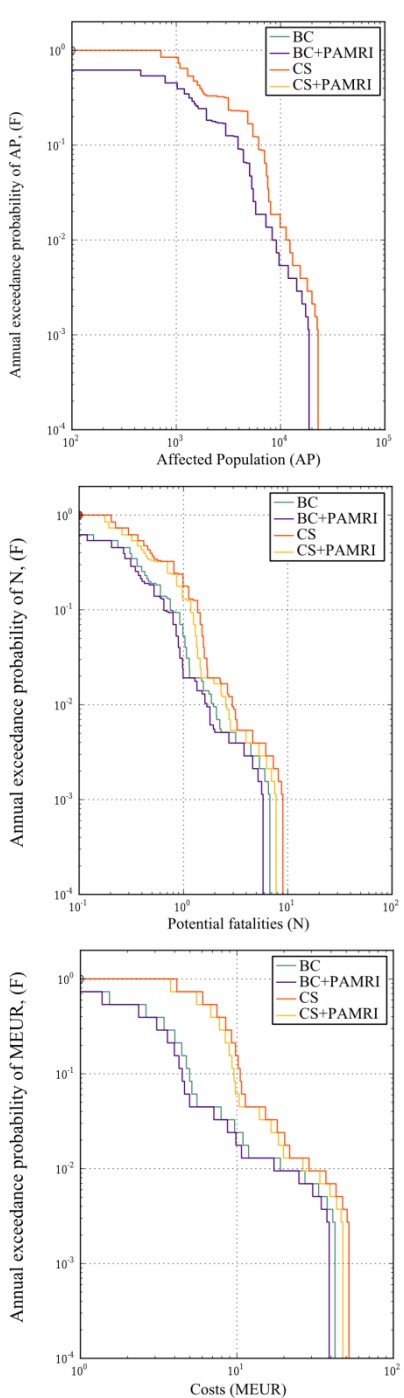

**Figure 7: Extract of F-AP, F-N and F-D curves for the case study: current situation (CS), base case (BC), and effect of local action plan (PAMRI).**





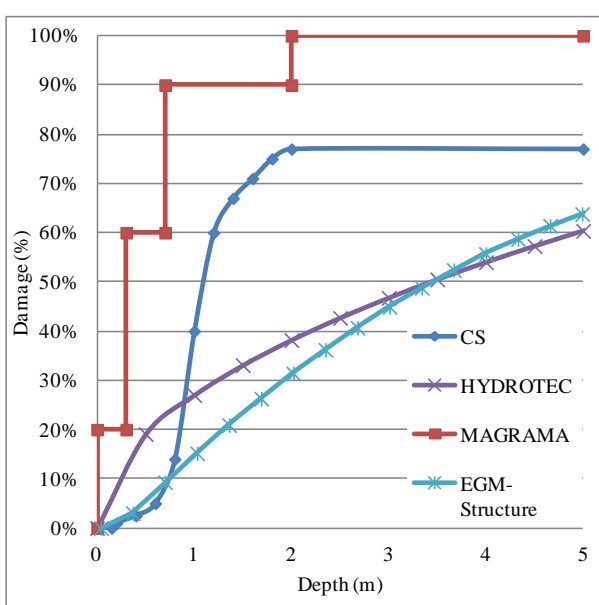

**Figure 8: Examples of depth-damage functions compared to case study analysis curve (CS).**