# Peer review of "Enhancing local action planning through quantitative flood risk analysis: a case study in Spain"

_Natural Hazards and Earth System Sciences, 2016_

## Referee Comment (RC1) · Anonymous Referee #1 · 8 Mar 2016

Reviewer Comments on the manuscript nhess-2016-065 Title: Enhancing local action planning through quantitative flood risk analysis: a case study in Spain.

General comments This paper is a "middle of the road" paper between a methodological paper and a policy oriented paper. It can have an important interest for the targeted readers (policy makers). However, at this stage, none are addressed carefully and the policy recommendations are too often disconnected from the results of the paper. I recommend to improve the relation between the methodological challenges of flood risk assessment and the policy orientation of the paper. I did some suggestions below and in the Specific comments.

The structure of the paper needs to be revised. Since there is no methodological contribution to the existing literature on risk assessment unless you did not highlight them sufficiently, then I would recommend to reorganize the structure of the paper and more precisely of section 3. Some subsections of section3 are made with 2 sentences in a total of 2/3 lines: this reveals a problem in your paper's structure. I would follow an overall classical structure: Method-Results-Discussion. This means to restructure the method section and the method (calibration) of the case study since you did not propose an original method. You should have only one method section (merge section 2 and 3.4) Related to these your Fig 3 is much more informative than your Fig1 and constitute your model/method. I would organize a method section around your model (described in fig3) and not around a generic method that you did not improve but replicate and which leads to some misunderstanding as noticed in the Specific comments

The policy oriented research questions (page3 line30) and the results are quite disconnected. The paper needs to improve this drastically. I mean that, choosing a quantitative hydro-eco model to assess flood risk and simulate risk management strategies is a very good research approach and is data intensive but it also means that the conclusions you derive from the model are supported and illustrated by your results (results+sensitivity sections). This is not the case with your policy recommendations. This weakens the interest of doing quantitative models.

A policy maker should be able to understand on which parameters of your model he/she could play to reduce the risk and which part of the model are more sensitive: Flows, AV, Y, stage functions,etc. This means to improve the readability of the paper, highlight the methodological points that have policy implications and also clearly and structurally present the parameters you use and those you use to illustrate the sensitivity of the results. The paper wold benefit of a summary table of the list of parameters, values, sources, model used.   Specific comments Introduction Page3 line 30-31: how your method helps to answer these questions is not clear. Why standardization is needed and which type of standardization is required to respect "Think globally, act locally" you are citing, ie to respect local properties with standards. This have to be discussed.

Section2 Method and the subsections You should use the Phase number in the subsection title. Ex. Review of available data→Phase II: review of available data. Also separate phase 7 and 8 in page 4. This helps to follow. But Given the irregular size of the description of these phases (some a definition other are models, some have 2 sentences only) consider another structure to present and develop these phases.

Page 5 lines 7-8. Subsection 2.3 Base Case. A current situation is not by convention a scenario. Then in the case study, page 10 Line 20 you refer to the Base Case as the scenario with dam. This is not coherent.

Page5 Line 10. What do you mean by plausible. Define, be more precise. What the flood defense sentence (line12-13) has to do with this section 2.4. Develop this.

Page 5 line 24 "Failure of a flood defense (e.g. a dam). Do you mean dam break? Figure1. You have 2 architecture models in the same figure. The indexes a and b only appear the in the caption, not in the figure. To be corrected.

Figure1: we understand that the economic damage D are estimated from the preceding step N which is the number of fatalities, meaning that your economic valuation is reduced to valuation of life lost. But them in the case study, we learn N and D are not connected. So Fig 1 can lead to a misleading lecture of your method/model. On the contrary Figure 3 which is not even mentioned in your text is more detailed and more informative for the reader. The literature has several illustrative flowcharts more informative than your Fig1 (or Fig 6) like : Messner and Meyer 2005 In: UFZ Discussion Papers 13/2005, Foudi et al. 2015 in Land Use Policy Pelling 2001 in Social Nature, or de Moel et al. 2015 in Mitig Adapt Strateg Glob Change

I recommend to correct Fig1 or think about the usefulness of this Figure for the reader (policy maker in your case). I believe that this architecture could be presented by inspiration of these examples in a way that the diagram reads easily. Also we cannot understand Fig1 without having to constantly refer to the text, this does not help. You could do it all in one single figure with text instead of letters. Also consider usefulness

of Fig 1 vs Fig3 when you restructure the paper and develop a unique methodological section.

Page 6 Line 21-30 Reorder the description of the architecture: Routing before NOL.

Page 6 Line 25: Gate operability. Define this for the reader. A policy maker that you target as reader may not be aware of this terminology

Page 6 Line 25: Routing. Define this for the reader.

Page 9 Line 8. What do you mean by tolerability. Explain, define

Section 3 Case study Consider to reduce (regroup) the number of subsections when you restructure the paper as explained in the General comments.

Page 10 Line 3 "now" When it started, when it is finished (planned to be finished). So, the reader can understand better that this is the policy scenario you simulate (one of the two).

Page 10 Line18 Line 20: Check the use of Current and Base. The benchmark (Base) is by convention the baseline case to which you compare a scenario. Correct the confusion this creates. The benchmark should be the no dam case (current as you call it), the other are scenarios.

Section 2.3 Consequence of estimation. Given that risk is the product of flood hazard and the negative consequences of flooding, you are mixing in this section exposure and vulnerability issues. But we miss some information as explain below:

Page 10 Line 11-14 Lines 31-34. This information is not understandable by the reader you target. Only expert that applied the DEFRA algorithm can understand it. You should explain sufficiently the algorithm of DEFRA so the area and people vulnerabilities are understandable. Also this way policy maker can understand on which parameters they could act to reduce the risk and how they could do it. This is an example of the points that could be developed in your discussion section so that you would improve

the connection between your model and the policy implications.

Page 10 Line 16. We miss information about Land uses in the case study. Any tables or comments would be grateful.

Page 10 Lines 15-22. We miss relevant information about the economic stage damage functions. I mean more than what is said in these lines. Present the stage damage function you use in the main results (those of section 3.5) then you could say that you do a sensitivity analysis of the stage damage function.

Page 10 Line 16. How do you do the asset values adjustment? Explain for the reader.

Page 10 Lines23-24 Reference is missing for these indirect costs. How do you select this 7%?

Page 10. Table 3. We should be able to read the table without having to come back to the text. Add the scenario definition in the Note of the table or rename them more explicitly for the reader.

Page 13 Line 6. Model section. Think about a restructuration with the section where you present the model. You should make reference to Fig3 somewhere. This fig presents your model, not a generic model as Fig1 (see earlier comment).

Page 13 Results. You should report the results of all your scenario in terms of expected risks as done in Table3 for the consequences estimation. We miss the results of the case where CS+Pampri.

Page 13 Line 28 "Figure 7shows...".This formulation is quite synthetic to present results!! You should guide the reader and develop your ideas.

Page 13 Section 3.6 I totally miss the interest/objective of this section. Do you really need a section for 1 sentence?

Page 14 Section 3.7.1. It is not clear how you perform this analysis. Why do you need the Aqueduc Global Analyser Database? Explain explicitly the parameter you've

played with and what are the expected consequences on the estimation of the risk of giving the 0 to the 1 year return period or to the 10 year. Discuss how others did in the literature and motivate why this is important (for the policy maker)

Page 14 line 10. Explain selected versus actual? Not clear

Page 14 Line17. Are the results obtained under what you call Current scenario or Base Case. Give the information.

Page 15 Line 14-16. Are the results obtained under what you call Current scenario or Base Case?. Give the information.

Page 15 Line 1. You are normally using the flow velocity to assess human health risks (DEFRA algorithm). Did you?

Page 15 Section 3.7.4. There is no simulation results. You say they will be presented in section 3.7.5.So what is the interest of this section as it is now? Consider to revise it to give relevant information for the sensitivity analysis. If you only discuss the sensitivity of the stage damage function, you could make reference to the literature of sensitivity analysis among others Moel et al. (2011, 2012); Saint-Geours et al. (2013), etc

Page 16 Line117-19 Are the results obtained under what you call Current scenario or Base Case? Give the information

Section Discussion. You should improve the discussion, selected the point you want to emphasize and really develop in a much better way than with bullet points where the interpretation is sometimes left to the reader.

Page 17 Line 9. Develop. You cannot expect the reader to interpret this.

Page 17 Line 11. What is your point? How do you jump for example from flood peak to population parameters in the uncertainty analysis. Develop your ideas.

Section 4.2 Local planning implications. This was a good intention but unfortunately the points you mention are disconnected from your results and this weakens the paper and

the interest to do a quantitative analysis. Use the sensitivity analysis and calibration of the model to orientate the policy maker towards the parameters he/she can play with to define prevention policies and measure their expected consequences. I refer to those parameters/elements that enter in the model and affect the risk management. The other parameters (those you use in your bullet points) have no quantified effect and deals with the large epistemic uncertainty of the flood risk assessment. I recommend to separate what your paper has shown (in the results and sensitivity) and its uncertainty related limitations, ie those parameters for which you cannot simulate the sensitivity by lack of knowledge.

Section 4.3 Recommendations. You already do recommendations in the section 4.2. Reorganize your sections. Also make sure that your recommendations are derived from your results. Example, bullet point 1: did your paper deals with coastal flooding or reveal (quantify) how important coastal flooding is? I don't think.

---

## Referee Comment (RC2) · Anonymous Referee #2 · 15 Apr 2016

In this paper a method for flood risk assessment at the local level is presented and tested in a municipality in Spain. The method includes all relevant parts of flood risk assessment (from hazard modelling to modelling different social and economic consequences), is able to show the impacts of different flood risk management measures and is tailored to the local needs and data availabilities. Therefore it is a valuable contribution to research on flood risk assessment & management in particular for the local level. However, the paper still needs to be improved substantially (I have read the comments from reviewer 1, and I agree to almost all of them – so I will not repeat all of them). My major comments are the following:

Objective of the paper: Currently there seem to be different aims of the paper: 1) To

present a framework for flood risk analysis and apply it to a city as an example for enhanced local flood risk management (p.1). I would agree to this objective. 2) "... to find answers to the following questions: (i) how local flood risk management strategies may benefit from risk analysis; and (ii) which are the current barriers to standardized local flood risk analysis?" (p.3). This would be more an institutional analysis, but I think the paper does not really address these questions, so I would revise these research questions and concentrate on the objective mentioned on page 1.

The statement on p 16: ("It is proposed as a standardized framework for enhancing local flood risk analysis at regional, national and pan-European scale." ) is for me definitely too ambitious. Proposing a standardized framework would require to carry out a detailed review of other flood risk assessment frameworks so far (e.g. DEFRA's FCD-PAG, recommendations from the Floodsite project, other national approach such as the damage scanner or the Planning Kit DPRD in the Netherlands, FLEMO in Germany,...) and to make clear how the new approach differs from these. But from my point of view it is not really the objective of the paper to propose a standardized framework for Europe, it is more to present a framework for Spain and to adjust it to local conditions (see point above). I would nevertheless recommend to add some more citations to other approaches.

Why is standardization necessary? My experience is that standardization in Europe is hardly possible due to different kind of available data, different regional circumstances etc. Isn't it more important to adjust FRA to local conditions (as you did in your study)? Please discuss why you think that standardization is required.

Own innovations: You are using and combining many already existing tools and approaches, but at least as far as I know two elements of your approach are quite novel: a) consideration of seasonality and b) FRA in highly regulated river systems. Unfortunately the latter is not applied in the case study, but nevertheless I would recommend to make more clear what your own new contributions to flood risk assessment are.

Risk assessment framework: The framework described in section 2 has many phases and there seem to be some redundancies (e.g. review of data and input data) so I would recommend to check if the steps of the framework could be simplified a little bit more. Furthermore, even the authors seem to have problems to follow these steps when they conduct the case study, i.e. in section 3 the structure is a little bit different. I would recommend to harmonize this with the steps in section 2 (or to merge both sections, as recommended by reviewer 1).

The final (and for decision support probably most important) phase, the evaluation and comparison of different risk management options seems to be a little bit under-developed so far. I would recommend to mention at least briefly decisions support approaches such as CBA and MCA. Furthermore it would be interesting for the reader to know what the current practice in Spain is regarding such approaches. Are they required by law as in the UK?

Please consider also my comments in the attached PDF-file.

Please also note the supplement to this comment:
http://www.nat-hazards-earth-syst-sci-discuss.net/nhess-2016-65/nhess-2016-65-RC2-supplement.pdf

**Supplement:**

[revised manuscript text omitted]

---

## Author Comment (AC1) · 7 Jun 2016

Journal:             NHESS
Title:               Enhancing local action planning through quantitative flood risk analysis: a case study in Spain
Author(s):           J. T. Castillo-Rodríguez et al.
MS No.:              nhess-2016-65
MS Type:             Research article
Special Issue:       Resilience and vulnerability assessments in natural hazards and risk analysis

**Note**

Author comments in **Bold.** The following notation is used: RC=Referee comment, AC=author comment, G=general comment, S=specific comment, P=page, and L=line (P and L refer to the first version of the manuscript).

**Response to General Comments from Anonymous Referee #1**

General comments:

RC1.G1. This paper is a "middle of the road" paper between a methodological paper and a policy oriented paper. It can have an important interest for the targeted readers (policy makers). However, at this stage, none are addressed carefully and the policy recommendations are too often disconnected from the results of the paper. I recommend to improve the relation between the methodological challenges of flood risk assessment and the policy orientation of the paper. I did some suggestions below and in the Specific comments.

**RC1.G1.AC: As suggested also by Referee 2 (RC2.S17), section 4.2 (P17) has been further improved to connect results with policy recommendations.**

RC1.G2.The structure of the paper needs to be revised. Since there is no methodological contribution to the existing literature on risk assessment unless you did not highlight them sufficiently, then I would recommend to reorganize the structure of the paper and more precisely of section 3. Some subsections of section3 are made with 2 sentences in a total of 2/3 lines: this reveals a problem in your paper's structure. I would follow an overall classical structure: Method-Results-Discussion. This means to restructure the method section and the method (calibration) of the case study since you did not propose an original method. You should have only one method section (merge section 2 and 3.4) Related to these your Fig 3 is much more informative than your Fig1 and constitute your model/method. I would organize a method section around your model (described in fig3) and not around a generic method that you did not improve but replicate and which leads to some misunderstanding as noticed in the Specific comments.

**RC1.G2.AC: As suggested, section 2 has been modified and the order of Fig.1 and 3 is changed to better explained the proposed process for flood risk analysis.**

RC1.G3. The policy oriented research questions (page3 line30) and the results are quite disconnected. The paper needs to improve this drastically. I mean that, choosing a quantitative hydro-eco model to assess flood risk and simulate risk management strategies is a very good research approach and is data intensive but it also means that the conclusions you derive from the model are supported and illustrated by your results (results + sensitivity sections). This is not the case with your policy recommendations. This weakens the interest of doing quantitative models.

**RC1.G3.AC: As suggested also by Referee 2 (RC2.S17), sections 4.2 and 4.3 (P17, now Section 5) have been improved to better explain how results from flood risk analysis may support policy strategies.**

RC1.G4. A policy maker should be able to understand on which parameters of your model he/she could play to reduce the risk and which part of the model are more sensitive: Flows, AV, Y, stage functions, etc. This means to improve the readability of the paper, highlight the methodological points that have policy implications and also clearly and structurally present the parameters you use

and those you use to illustrate the sensitivity of the results. The paper would benefit of a summary table of the list of parameters, values, sources, model used.

**RC1.G4.AC: Thank you for this suggestion. We have included a summary table (Table 1) with main parameters, values, sources of information… to improve readability of the article.**

**Response to Specific Comments from Anonymous Referee #1**

**Specific comments:**

RC1.S1.     Introduction Page3 line 30-31: how your method helps to answer these questions is not clear. Why standardization is needed and which type of standardization is required to respect "Think globally, act locally" you are citing, ie to respect local properties with standards. This have to be discussed.

**RC1.S1.AC: Thank you for this comment. As also suggested by referee 2, emphasis is now on the main objective and this paragraph has been changed and now reads as follows:**

**"This paper aims to present a framework for local flood risk analysis and its application to a real case to show how local flood risk management strategies may benefit from risk analysis."  Standardization at pan-European scale is not the main scope of the presented framework, but presenting a robust framework for local flood risk analysis.**

RC1.S2.     Section2 Method and the subsections. You should use the Phase number in the subsection title. Ex. Review of available data-> Phase II: review of available data. Also separate phase 7 and 8 in page 4. This helps to follow. But Given the irregular size of the description of these phases (some a definition other are models, some have 2 sentences only) consider another structure to present and develop these phases.

**RC1.S2.AC: Section 2 ("Method"→"Approach") and 3 ("Case study") have been changed to provide a more clear structure.**

RC1.S3.     Page 5 lines 7-8. Subsection 2.3 Base Case. A current situation is not by convention a scenario. Then in the case study, page 10 Line 20 you refer to the Base Case as the scenario with dam. This is not coherent.

**RC1.S3.AC: Scenario definition is changed based on suggestions from both referees and now the benchmarking scenario refers to the situation without structural measures (current situation).**

RC1.S4.     Page5 Line 10. What do you mean by plausible. Define, be more precise. What the flood defense sentence (line12-13) has to do with this section 2.4. Develop this.

**RC1.S4.AC: Corrected**

RC1.S5.     Page 5 line 24 "Failure of a flood defense (e.g. a dam). Do you mean dam break?

**RC1.S5.AC: It includes potential break of existing flood defense (e.g. dam or levees).**

RC1.S6.     Figure1. You have 2 architecture models in the same figure. The indexes a and b only appear the in the caption, not in the figure. To be corrected.

**RC1.S6.AC: Corrected**

RC1.S7.     Figure1: we understand that the economic damage D are estimated from the preceding step N which is the number of fatalities, meaning that your economic valuation is reduced to valuation of life lost. But them in the case study, we learn N and D are not connected. So Fig 1 can lead to a misleading lecture of your method/model. On the contrary Figure 3 which is not even mentioned in your text is more detailed and more informative for the reader. The literature has several illustrative flowcharts more informative than your Fig1 (or Fig 6) like : Messner and Meyer 2005 In: UFZ Discussion Papers 13/2005, Foudi et al. 2015 in

Land Use Policy Pelling 2001 in Social Nature, or de Moel et al. 2015 in Mitig Adapt Strateg Glob Change.

**RC1.S7.AC: Figure 1 is corrected and Figure 3 appears now first. Further explanation is provided to describe N and D.**

**Figure 1 and 6 (figures 3 and 6 in the revised version) represent the risk model architecture (nodes and connectors that compose the proposed generic architecture and the case-study specific architecture). However, these schemes do not aim at describing all flood risk components but the structure of nodes integrating information regarding probabilities and consequences.**

**For a more clear explanation, both figures are now shown after Figure 3 (flowchart describing the flood risk analysis process).**

RC1.S8.        I recommend to correct Fig1 or think about the usefulness of this Figure for the reader (policy maker in your case). I believe that this architecture could be presented by inspiration of these examples in a way that the diagram reads easily. Also we cannot understand Fig1 without having to constantly refer to the text, this does not help. You could do it all in one single figure with text instead of letters. Also consider usefulness of Fig 1 vs Fig3 when you restructure the paper and develop a unique methodological section.

**RC1.S8.AC: Figure 1 (Figure 3 in the revised version) now includes information in each node.**

RC1.S9.        Page 6 Line 21-30 Reorder the description of the architecture: Routing before NOL.

**RC1.S9.AC: Figure 1 (Figure 3 in the revised version) has been corrected.**

RC1.S10.        Page 6 Line 25: Gate operability. Define this for the reader. A policy maker that you target as reader may not be aware of this terminology.

**RC1.S10.AC: Corrected**.

RC1.S11.        Page 6 Line 25: Routing. Define this for the reader.

**RC1.S11.AC: Further definition is added.**

RC1.S12.        Page 9 Line 8. What do you mean by tolerability. Explain, define.

**RC1.S12.AC: Further definition is added and a reference to generalized tolerability criteria.**

RC1.S13.        Section 3 Case study Consider to reduce (regroup) the number of subsections when you restructure the paper as explained in the General comments.

**RC1.S13.AC: Sections 2 and 3 have been changed.**

RC1.S14.        Page 10 Line 3 "now" When it started, when it is finished (planned to be finished). So, the reader can understand better that this is the policy scenario you simulate (one of the two).

**RC1.S14.AC: Further information on dam construction stage is provided in the revised version.**

RC1.S15.        Page 10 Line18 Line 20: Check the use of Current and Base. The benchmark (Base) is by convention the baseline case to which you compare a scenario. Correct the confusion this creates. The benchmark should be the no dam case (current as you call it), the other are scenarios.

**RC1.S15.AC: This is corrected in the revised version.**

Section 2.3 Consequence of estimation. Given that risk is the product of flood hazard and the negative consequences of flooding, you are mixing in this section exposure and vulnerability issues. But we miss some information as explain below:

RC1.S16.     Page 12 Line 11-14 Lines 31-34. This information is not understandable by the reader you target. Only expert that applied the DEFRA algorithm can understand it. You should explain sufficiently the algorithm of DEFRA so the area and people vulnerabilities are understandable. Also this way policy maker can understand on which parameters they could act to reduce the risk and how they could do it. This is an example of the points that could be developed in your discussion section so that you would improve the connection between your model and the policy implications.

**RC1.S16.AC: Further description of the DEFRA methodology is provided. The new paragraph reads as follows "Estimation of potential life-loss is based on the method proposed by DEFRA (DEFRA, 2006). The number of fatalities is a function of the number of injuries and the hazard rating, where the number of injuries is estimated by combining the following factors:**

— **Number of people within the hazard zone.**
— **Hazard rating: factor which combines flood characteristics (flood depth, flow velocity and debris factor).**
— **Area vulnerability: function of effectiveness of flood warning, speed of onset of flooding and nature of area (including types of buildings); and**
— **People vulnerability: function of presence of people who are very old and/or infirm/disabled/long-term sick."**

RC1.S17.     Page 12 Line 16. We miss information about Land uses in the case study. Any tables or comments would be grateful.

**RC1.S17.AC: Land use types are included in Table 5 in the revised version.**

RC1.S18.     Page 12 Lines 15-22. We miss relevant information about the economic stage damage functions. I mean more than what is said in these lines. Present the stage damage function you use in the main results (those of section 3.5) then you could say that you do a sensitivity analysis of the stage damage function.

**RC1.S18.AC: The stage damage function is added.**

RC1.S19.     Page 12 Line 16. How do you do the asset values adjustment? Explain for the reader.

**RC1.S19.AC: Asset values used for the analysis are those used by regional planning for different land use types. Reference is added (in Spanish) and values are provided in Table 5.**

RC1.S20.     Page 12 Lines23-24 Reference is missing for these indirect costs. How do you select this 7%?

**RC1.S20.AC: This value was estimated in a previous study conducted by UPV for the regional government. Reference is included (in Spanish).**

RC1.S21.     Page 12. Table 3. We should be able to read the table without having to come back to the text. Add the scenario definition in the Note of the table or rename them more explicitly for the reader.

**RC1.S22.AC: Content is further described.**

RC1.S22.     Page 13 Line 6. Model section. Think about a restructuration with the section where you present the model. You should make reference to Fig3 somewhere. This fig presents your model, not a generic model as Fig1 (see earlier comment).

**RC1.S22.AC: Section 3 has been restructured, following phases described in Section 2.**

RC1.S23.     Page 13 Results. You should report the results of all your scenario in terms of expected risks as done in Table3 for the consequences estimation. We miss the results of the case where CS+PAMRI.

**RC1.S23.AC: Corrected.**

RC1.S24.     Page 13 Line 28 "Figure 7 shows…:".This formulation is quite synthetic to present results!! You should guide the reader and develop your ideas.

**RC1.S24.AC: Results represented in Figure 7 are further described. This paragraph now reads as follows:**

"**Figure 7 shows F-AP, F-N and F-D curves for all scenarios. The first graph depicts the cumulative annual exceedance probability (F) of each level of potential affected population (AP). Results show that there is a probability of $10^{-2}$ of exceeding 8,300 affected people due to flooding for the scenario with structural measures. This value is higher when considering the current situation, with approx. 11,300 affected people for the same probability. The second graph depicts the cumulative annual exceedance probability (F) of each level of potential fatalities (N). Results show that there is a probability of $10^{-2}$ of exceeding 3 fatalities for the current situation (Scenario 0). This value decreases after implementing structural measures (Scenario 1) to approx. 2 and up to 1.6 for combined structural and non-structural measures (Scenario 3). The third graph shows potential economic damages (D) with a probability of $10^{-2}$ of exceeding 28 M€ for the current situation (Scenario 0). This value might decrease up to approx.17 M€ after implementing combined structural and non-structural measures (Scenario 3).**"

RC1.S25.     Page 13 Section 3.6 I totally miss the interest/objective of this section. Do you really need a section for 1 sentence?

**RC1.S25.AC: Subsection is not used. Risk maps are included as supplementary material to show how risk outcomes can be represented.**

RC1.S26.     Page 14 Section 3.7.1. It is not clear how you perform this analysis. Why do you need the Aqueduc Global Analyser Database? Explain explicitly the parameter you've played with and what are the expected consequences on the estimation of the risk of giving the 0 to the 1 year return period or to the 10 year. Discuss how others did in the literature and motivate why this is important (for the policy maker).

**RC1.S26.AC: Further description is provided. Input data on flood consequences is modified by establishing a different level of protection (i.e. no flood consequences are assumed for flood events with return periods below that level). The benchmarking scenario was analyzed assuming a protection level related to a flood event with a 1-yr return period.**

**The Aqueduct Global Analyser Database is used as an example to allow comparison with available regional analysis.**

RC1.S27.     Page 14 line 10. Explain selected versus actual? Not clear.

**RC1.S27.AC: "Selected" refers to the assumption included in the case study analysis. "Actual" refers to the real, not known, level of protection. This sentences is corrected.**

RC1.S28.     Page 14 Line17. Are the results obtained under what you call Current scenario or Base Case. Give the information.

**RC1.S28.AC: Results refer to the current situation, corrected.**

RC1.S29.     Page 15 Line 14-16. Are the results obtained under what you call Current scenario or Base Case?. Give the information.

**RC1.S29.AC: Results refer to the current situation, corrected.**

RC1.S30.     Page 16 Line 1. You are normally using the flow velocity to assess human health risks (DEFRA algorithm). Did you?

**RC1.S30.AC: This line refered to economic damages. Flow velocity is used on consequence estimation in terms of life-loss but stage-damage functions do not consider flow velocity.**

RC1.S31.    Page 15 Section 3.7.4. There is no simulation results. You say they will be presented in section 3.7.5.So what is the interest of this section as it is now? Consider to revise it to give relevant information for the sensitivity analysis. If you only discuss the sensitivity of the stage damage function, you could make reference to the literature of sensitivity analysis among others Moel et al. (2011, 2012); Saint-Geours et al. (2013), etc.

**RC1.S31.AC: Results were provided in section 3.7.5. since MAGRAMA also proposes different asset values to be used along with stage-damage functions, then results,  from modifying both asset values and damage function, are presented in section 3.7.5.**

RC1.S32.    Page 16 Line17-19 Are the results obtained under what you call Current scenario or Base Case? Give the information.

**RC1.S32.AC: The current scenario is used for comparison (included in brackets).**

RC1.S33.    Section Discussion. You should improve the discussion, selected the point you want to emphasize and really develop in a much better way than with bullet points where the interpretation is sometimes left to the reader.

**RC1.S33.AC: As suggested, this section has been further developed.**

RC1.S34.    Page 17 Line 9. Develop. You cannot expect the reader to interpret this.

**RC1.S34.AC: As suggested, this point has been further developed.**

RC1.S35.    Page 17 Line 11. What is your point? How do you jump for example from flood peak to population parameters in the uncertainty analysis. Develop your ideas.

**RC1.S35.AC: As suggested, this point has been further developed.**

RC1.S36.    Section 4.2 Local planning implications. This was a good intention but unfortunately the points you mention are disconnected from your results and this weakens the paper and the interest to do a quantitative analysis. Use the sensitivity analysis and calibration of the model to orientate the policy maker towards the parameters he/she can play with to define prevention policies and measure their expected consequences. I refer to those parameters/elements that enter in the model and affect the risk management. The other parameters (those you use in your bullet points) have no quantified effect and deals with the large epistemic uncertainty of the flood risk assessment. I recommend to separate what your paper has shown (in the results and sensitivity) and its uncertainty related limitations, ie those parameters for which you cannot simulate the sensitivity by lack of knowledge.

**RC1.S36.AC: Sections 4.2. and 4.3 are further developed to incorporate connections to previous sections (mainly section 4, sensitivity analysis) and results from QRA with local action planning.**

RC1.S37.    Section 4.3 Recommendations. You already do recommendations in the section 4.2. Reorganize your sections. Also make sure that your recommendations are derived from your results. Example, bullet point 1: did your paper deals with coastal flooding or reveal (quantify) how important coastal flooding is? I don't think.

**RC1.S37.AC: Coastal flooding is not included in the analysis due to lack of information but it would be desirable to analyze risk from coastal flooding in future upgrades for this case study.**

**Response to General Comments from Anonymous Referee #2**

In this paper a method for flood risk assessment at the local level is presented and tested in a municipality in Spain. The method includes all relevant parts of flood risk assessment (from hazard modelling to modelling different social and economic consequences), is able to show the impacts of different flood risk management measures and is tailored to the local needs and data availabilities. Therefore it is a valuable contribution to research on flood risk assessment & management in particular for the local level.

However, the paper still needs to be improved substantially (I have read the comments from reviewer 1, and I agree to almost all of them – so I will not repeat all of them).

My major comments are the following:

RC2.G1. Objective of the paper: Currently there seem to be different aims of the paper: 1) To present a framework for flood risk analysis and apply it to a city as an example for enhanced local flood risk management (p.1). I would agree to this objective. 2) "… to find answers to the following questions: (i) how local flood risk management strategies may benefit from risk analysis; and (ii) which are the current barriers to standardized local flood risk analysis?" (p.3). This would be more an institutional analysis, but I think the paper does not really address these questions, so I would revise these research questions and concentrate on the objective mentioned on page 1.

**RC2.G1.AC: As mentioned, the main objective of this paper is to present a framework for flood risk analysis and its application to a city. Section 4 (now 5) has been further described to provide answers to the questions included in page 3 (additional objectives, derived from research work conducted).**

RC2.G2. The statement on p.16: ("It is proposed as a standardized framework for enhancing local flood risk analysis at regional, national and pan-European scale." ) is for me definitely too ambitious. Proposing a standardized framework would require to carry out a detailed review of other flood risk assessment frameworks so far (e.g. DEFRA' FCDPAG, recommendations from the Floodsite project, other national approach such as the damage scanner or the Planning Kit DPRD in the Netherlands, FLEMO in Germany,…) and to make clear how the new approach differs from these. But from my point of view it is not really the objective of the paper to propose a standardized framework for Europe, it is more to present a framework for Spain and to adjust it to local conditions (see point above). I would nevertheless recommend to add some more citations to other approaches.

**RC2.G2.AC: Thank you for pointing this out. We have clarified that the proposed framework is potentially replicable to other locations in Europe, but not aiming at providing a pan-European analysis. As suggested, this statement has been changed and examples to other approaches are included in the revised version.**

RC2.G3. Why is standardization necessary? My experience is that standardization in Europe is hardly possible due to different kind of available data, different regional circumstances, etc. Isn't it more important to adjust FRA to local conditions (as you did in your study)? Please discuss why you think that standardization is required.

**RC2.G3.AC: We agree with the reviewer on the difficulties to develop standardize flood risk analyses among different countries in Europe. However, we think that a certain level of standardization should be achieved, at least for urban areas in the same region or country, to ensure robust flood risk analyses. Adaptation to local conditions is always required but based on a robust flood risk analysis process when analyzing flood risk at different locations, as pointed out in section 4.**

RC2.G4. Own innovations: You are using and combining many already existing tools and approaches, but at least as far as I know two elements of your approach are quite novel: a) consideration of seasonality and b) FRA in highly regulated river systems. Unfortunately the latter is not applied in the case study, but nevertheless I would recommend to make more clear what your own new contributions to flood risk assessment are.

**RC2.G4.AC: Main innovations of this paper are: 1) a robust framework for quantitative and comprehensive local flood risk analysis using event tree modeling and GIS-based information, 2) an**

**application to a city by analyzing different scenarios with structural and non-structural measures for flood risk reduction and 3) an approach for risk-informed local action planning against flood risk. The revised version of sections 1.3, 4.2 and 4.3 now clarifies these points.**

RC2.G5. Risk assessment framework: The framework described in section 2 has many phases and there seem to be some redundancies (e.g. review of data and input data) so I would recommend to check if the steps of the framework could be simplified a little bit more. Furthermore, even the authors seem to have problems to follow these steps when they conduct the case study, i.e. in section 3 the structure is a little bit different. I would recommend to harmonize this with the steps in section 2 (or to merge both sections, as recommended by reviewer 1).

**RC2.G5.AC: As suggested, the structure of sections 2 and 3 is modified to provide a more clear description of the method and case study application.**

RC2.G6. The final (and for decision support probably most important) phase, the evaluation and comparison of different risk management options seems to be a little bit underdeveloped so far. I would recommend to mention at least briefly decisions support approaches such as CBA and MCA. Furthermore it would be interesting for the reader to know what the current practice in Spain is regarding such approaches. Are they required by law as in the UK?

**RC2.G6.AC: Section "Study of risk reduction measures" has been completed with content regarding decision-support approaches. In addition, section 3 now includes the evaluation of proposed risk reduction measures based on the ACSLS indicator.**

**Response to Specific Comments from Anonymous Referee #2**

From comments in supplementary material from Referee 2 (PDF file), we here include answers and corresponding changes in the revised version of the manuscript:

| Comment | P/L | Answer |
|---------|-----|--------|
| RC2.S1 | 2/5 | Corrected, the paragraph reads "The trend will probably continue to rise as floods and storms are expected to become more frequent and severe in Europe in the future (UNISDR, 2009) due to climate and socioeconomic change." |
| RC2.S2 | 2/30 | Corrected, added examples of detailed FRA in Europe. |
| RC2.S3 | 3/25 | Added definitions for risk components. |
| RC2.S4 | 4/5 | Corrected. |
| RC2.S5 | 5/1 | Section 2 has been changed and sub-chapters are not used. |
| RC2.S6 | 7/10 | Further explanation is provided on the use of software tools. |
| RC2.S7 | 7/23 | Third-level hierarchy is not used in this point as suggested. |
| RC2.S8 | 8/27 | Explanation on why environmental risks are not included is provided, along with examples of FRA as suggested. |
| RC2.S9 | 9/12 | Risk thresholds proposed by USBR refer to population downstream a dam. |
| RC2.S10 | 9/15 | Further explanation on decision support methods is included. |
| RC2.S11 | 11/6 | As suggested, only the current situation is used for benchmarking. |
| RC2.S12 | 13/6 | Section 3.4 refers to Risk Model. |
| RC2.S13 | 16/17 | RC: I do not know how the two sources derived these land-use related values, but one explanation why MAGRAMA have higher values might be that they use replacement costs, which is often an overestimation of the time value of assets (see Messner et al 2007: recommendations of flood damage |

|  |  | evaluation). |
|  |  | In general the use of standardized asset values is an approach which is more related to macro-scale flood damage evaluations, which does not really fit to your micro-scale approach. I.e. it would be better: |
|  |  | a) derive asset values from official statistics for the municipality and break them down to the corresponding land use categories or |
|  |  | b) estimate building-specific asset values (see also Messner et al. 2007 for recommendations on approaches for meso and micro-scale assessments). |
|  |  | It is understood that this maybe cannot be changed anymore but it should be mentioned in the discussion, that it this would be an important step to improve the level of detail of the overall results. |
|  |  | **AC: As stated in page 16, rates proposed by MAGRAMA include replacement costs for infrastructure, content and vehicles, not included by regional studies. As suggested, mention is included to the use of land-use related values instead of asset or building-specific.** |
| RC2.S14 | 16/30 | Sentence has been changed to better explain that the risk assessment framework is transferable to other local applications in Europe. |
| RC2.S15 | 17/10 | The estimation of asset values is mentioned as suggested. |
| RC2.S16 | 17/16 | Corrected to "in Spain". |
| RC2.S17 | 17/18 | How are the following recommendations related to your results in the case study (or experiences of the application of the risk assessment framework)? |
|  |  | **AC: Further explanation is provided on how results support recommendations, connecting to content in Section 4 (revised version).** |
|  | 18/6 | Corrected. |
| RC2.S18 | 18/8 | At the beginning of the article you stated that local authorities currently do not have the know-how, data etc. to conduct risk assessments. Please briefly discuss how this can be improved. Is it realistic that local authorities conduct detailed quantitative risk analysis on their own? Or do they have the resources to pay consultancies for that? |
|  |  | **AC: Further explanation is provided on how local authorities may conduct QRA.** |
| RC2.S19 | 18/17 | In some countries such guidance already exist, e.g. in England and Wales with the "flood and coastal defence appraisal guidance" from DEFRA 1999. |
|  |  | **AC: As suggested, reference to FR guidance in England and Wales is provided. The paragraph now reads as follows:** |
|  |  | **Quantification of societal and economic flood risk is not required by current legislation in Spain and is relatively novel in local flood risk management as a result a lack of guidance, standardized methods or tools for local flood risk analysis. Examples can be found in other countries such as in England and Wales (Hall et al., 2003), but are still scarcely applied in Spain.** |

[revised manuscript text omitted]

Despite recent legislation and work conducted on flood risk management, there is still a need for local flood risk analyses to complete those developed at regional scale, to inform action planning and to better orientate risk reduction actions at urban scale. In most cases, despite some exemptions found in the literature such as guidance and examples of micro-scale flood
30   risk assessment carried out e.g. in England and Wales (Penning-Rowsell et al., 2013), there is a lack of applications of risk analysis techniques at local scale or the required level of detail to support decision-making on local flood risk reduction and planning.

**1.2 The challenge**

Flood risk management has acquired an important role since the European Floods Directive (Directive 2007/60/EC) and global strategies for flood risk reduction have evolved from focusing mainly on reducing the hazard (structural measures) to more holistic approaches including the combination of both hazard and impact mitigation.

5 Different approaches for flood risk analysis can be found in the literature, including societal (Jonkman et al., 2008) and economic risks (Merz et al., 2010), and ranging from local (Marcotullio and McGranahan, 2006) to global scale (Winsemius et al., 2013).

'Think globally, act locally', the famous phrase attributed to René Dubos during the UN Conference on the Human Environment in 1972, emphasizes the importance of scale in dealing with environmental challenges. Unique physical,

10 climatic, and cultural conditions appear at local scale and site-specific flood risk management is needed.

In the Valencian region, only 18 out of 136 local action plans for flood risk management have been developed and approved up to date. Despite the publication of some recommendations by civil protection on how to perform these plans, local authorities do not have the information, know-how or experience on the required flood risk analyses to be developed.

The city of Oliva, located in the Eastern coast of Spain, belongs to the group of municipalities within medium to high flood

15 risk levels. Located 70 km from Valencia, Oliva is affected by pluvial, river and coastal flooding and it is characterized by a complex and wide-ranging geography (e.g. hills up to 460 m.a.s.l., plains, coastal areas and wetlands). In addition, there is high seasonal variation in population (with 27,127 and 55,174 inhabitants of resident and seasonal population, respectively, distributed across 60.1 km²).

After 28 years of the largest flood event in modern times in Oliva (accounting for the highest recorded rainfall rate at the

20 Iberian Peninsula with 817 mm in 24 hours), local authorities face the challenge of mitigating flood risk through the development and implementation of a local action plan (as required by regional legislation), in line with other existent and ongoing structural measures for flood risk reduction.

Local authorities are aware of the need for improved flood risk management strategies and quantitative flood risk analysis arises as a helpful tool to support management actions and strategies.

**25 1.3 The opportunity**

Flood risk is commonly expressed in terms of expected annual damage (in terms of potential affected population, number of fatalities or economic damage), obtained from the combination of three key components: flooding probability, exposure determinants and vulnerability of receptors (Klijn et al., 2015). Generally, risk is conceptualized as the multiplication of flood probability and consequences. In this paper, we propose flood risk analysis through the use of risk models, capable of

30 estimating annual risk for different scenarios and performed for a real case study, based on results from flood hazard characterization and consequence estimations. This paper aims to present a framework for local flood risk analysis and its application to a real case to show how local flood risk management strategies may benefit from risk analysis. This paper

analyses the city of Oliva (Spain) as an example. Although flood hazard mapping is available, a quantitative flood risk analysis had never been performed. The presented study is being used as a basis for developing a local action plan against flood risk. This 'science for policy' paradigm can be considered as a 'lighthouse' example for other cities in Spain that are required to develop their corresponding plans.

5  The study shows how flood probability, exposure and vulnerability analyses provide valuable information for the development of a local action plan against flooding, for example by characterizing the impact on risk of improved warning systems and public education campaigns.

**2 Approach**

In this section, the applied framework (including tools and methods) for flood risk analysis is described. This framework for
10  flood risk analysis is based on the method proposed by (Escuder-Bueno et al., 2012), through the use of a risk model which incorporates all information regarding loads, system response and flood consequences, and adapted to integrate GIS data into risk modelling. Figure 1 shows the flowchart summarizing data, methods and tools within the presented framework.

Potential applications include local flood risk assessments such as those required by regional and national legislation in Spain after the 2007 European Floods Directive.

15  The steps of the proposed framework include:
- Phase I: Scope of the case study
- Phase II: Review of available data
- Phase III: Study of the system situation: definition of the Base Case
- Phase IV: Flood events to be analyzed
20  - Phase V: Risk model architecture
- Phase VI: Input data for the risk model
- Phase VII: Risk calculation
- Phase VIII: Risk representation
- Phase IX: Risk evaluation
25  - Phase X: Study of risk reduction measures

**2.1 Phase I: Scope of the case study**

The proposed framework aims at estimating flood risk in urban areas in terms of affected population, potential injuries, fatalities, and economic costs resulting from damage to assets and infrastructure. It can be applied to analyze existent risk or to compare different scenarios to evaluate the impact of risk reduction measures.

30  The level of detail of the required analysis will depend on the scope and scale of decisions for flood risk management.

**2.2 Phase II: Review of available data**

Information on hydrologic studies, hydraulic modelling, flood defence response, population and land use data is required for characterizing loads, system response and estimating consequences from flooding.

GIS data on flood characteristics (e.g. flood depth, velocity, flooded area, etc.) and population and land uses is required to apply the procedure proposed in Fig.1. In recent  years, more detailed and up-to-date GIS-based data is available.

**2.3 Phase III: Study of the system situation: definition of the Base Case**

The Base Case corresponds with the benchmark scenario. This scenario is used for analyzing the impact of risk reduction measures.

**2.4 Phase IV: Flood events to be analyzed**

The range of all potential flood events should be considered, obtained from hydrologic studies, and analyzed through hydraulic simulations to characterize system response and flood characteristics.

Flood defence reliability should be incorporated, when possible, into hydraulic modelling to analyze the existent protection level and the impact on flood characteristics of their performance (failure and non-failure cases of flood protection infrastructure).

This range will be divided into intervals, as shown in Fig.2, to incorporate data on flood hazard probabilities into the risk model performed in Phase V. Each flood event interval is characterized by a representative annual exceedance probability (AEP).

**2.5 Phase V: Risk model architecture**

The use of risk models provides a logic and mathematically rigorous framework for compiling information of the system to estimate flood risk (Castillo-Rodriguez et al., 2014).

The risk model can be represented by an influence diagram composed by nodes and connectors (Serrano-Lombillo et al., 2011). Nodes include information on loads (e.g. annualized probabilities of flood events), system response (failure probabilities of flood defence infrastructures, e.g. dam or levee breach) or consequences (e.g. results from consequence estimations in terms of affected population or economic damages).

In this paper, two generic schemes for defining the risk model architecture are proposed and shown in Fig.3. The first scheme (model "a") can be used for analyzing flood risk for urban areas affected by river flooding from non-regulated systems. The second scheme (model "b") should be used if potential failure of a flood defence (e.g. a dam) is incorporated into the analysis.

These two generic influence diagrams are an adapted version of those proposed by (Castillo-Rodriguez et al., 2014). These schemes allow to include the analysis of societal risk in terms of affected population, potential injuries, fatalities and economic costs due to damages from flooding (assets, infrastructure and services).

For the first influence diagram (model "a"), proposed to analyze flood risk in non-regulated systems, the following nodes are considered:

— Moment: this node includes information on probabilities for different time periods during the day (i.e. the probability of being during the day or at night). It can be used to later incorporate daily variability on potential consequences (e.g. affected population in industrial areas might change depending on the moment of the day).

— Season: this node includes information on probabilities for different seasonal periods during the year (i.e. the probability of being in summer or winter season). It can be used to later incorporate seasonal variability on potential consequences (e.g. affected population in urban areas might change if resident or potential population during summer is considered).

— Flood events: this node includes information on probabilities for different flood events. A range of flood events is established, defined by minimum and maximum return periods. This range is divided into a number of intervals (e.g. 10, 20, etc.). Figure 2 shows how the range of plausible flood events is divided into intervals for risk calculations. These intervals are equally spaced in logarithmic scale along the given range of return periods. Each interval is represented by an annual exceedance probability (AEP), obtained by deducting AEP values of low and high interval limits. The example shows a range from 1-yr to 1000-yr flood events into 10 intervals. An additional interval is added to include flood events that exceed the 1,000-yr return period.

— System response: this node includes information on system response (e.g. peak flow river discharges).

— Affected population, number of injured people, potential fatalities and economic costs: these nodes include information on consequence estimation in terms of affected population (AP), injuries (NI), fatalities (N) and economic damages in the urban area (D), respectively. Estimations for different flood events are obtained and incorporated into the risk model in each node.

For the second influence diagram (model "b"), proposed to analyze flood risk in a regulated river system with a dam, the following nodes are considered:

— Moment, Season, Flood events: these nodes are equivalent to the aforementioned described for the first influence diagram.

— Normal Operating Level (NOL): this node includes the water level at the reservoir in normal situation. For simplicity, it is assumed that this level is constant.

— Gate operability: this node includes probabilities for each possible combination of gate operability (number of gates functioning correctly for flood routing when the flood arrives) for dams with controlled outlet works.

— Routing: refers to results from the technique used to estimate evolution of water levels at river course and reservoirs during the flood event, based on initial conditions (water level when the flood arrives). Results from flood routing

are included in this node for each flood event and gate operability combination. Two outcomes from flood routing analysis are required: the maximum water pool level at the reservoir and resulting peak flow discharge through outlet works for each combination.

    — System response: for each load combination (represented by a maximum water pool level from flood routing), this node is used to consider two possible situations: failure and non-failure of the flood defence system, with complementary conditional probabilities of occurrence for each load combination. Hence, two branches emerge from this node to consider both options.

    — Failure and non-failure hydrographs: these nodes include information on peak flow discharges resulting from flood defense failure or non-failure cases (i.e. peak flow discharges from flood routing).

— Affected population, number of injured people, potential fatalities and economic costs: these nodes include information on consequence estimation in terms of affected population (AP), injuries (NI), fatalities (N) and economic damages in the urban area (D) for flood events resulting from flood defense failure (upper branch) and non-failure cases (lower branch).

**2.6 Phase VI: Input data for the risk model**

A GIS-based tool is proposed for input data processing. The tool, named gvSIG Desktop ([www.gvsig.com](www.gvsig.com)), is an open source software, GNU / GPL license, with free use, distribution, study and improvement. Recently, gvSIG has been graduated as an OSGeo project (Open Source Geospatial Foundation). This GIS software tool was first developed by the regional government of the Valencian Autonomous Region (to be widely implemented in their regional and local systems) and now is further developed and promoted by the gvSIG Association.

The use of other available GIS tools can be applied within this framework (e.g. qGIS). In this paper, gvSIG has been applied since it is being used by local governments in Spain.

In this paper, the procedure shown in Fig.1 is proposed to integrate GIS data into the risk model in Phase VII. This procedure shows the required steps to estimate flood consequences and to provide input data for the risk model in terms of affected population, potential injuries and fatalities and damage costs at local scale. This GIS-based procedure aims at boosting implementation of risk-informed local action plans through standardized consequence estimation and risk calculation.

The information required includes:

    — Hydrological and hydraulic modeling. Flood characteristics should be estimated for each cell on the map representing the study area for different floods (a range of flood events with return periods up to, at least, 500- year is recommended). Two maps are required showing inundation depths and flow velocities for each cell.

    — Consequence estimation. Several types of consequences per cell on the map are obtained. The impacts are then aggregated at municipality scale. The impacts include population exposed to flooding, injuries, potential fatalities and economic damages.

- o Affected population. Affected population should be obtained using census data (resident and seasonal population) and information on occupancy rates in hotels, campsites, etc.
- o Life-loss estimation. The life-loss estimation method proposed by MAGRAMA (Spanish Ministry of Agriculture, Food and Environment) for developing risk analysis at river basin scale is used. This method is based on the methodology proposed by DEFRA (Department for Environment, Food and Rural Affairs). Recent flood risk analyses have been conducted in Spain by applying this methodology, as for example in the Ebro River Basin (PREEMPT project "Policy-relevant assessment of socio-economic effects of droughts and floods"). For a detailed description on the method for estimating potential fatalities, the reader is referred to Wallingford et al (2006).
- o Economic damage estimation. It is based on the method used in PATRICOVA (Generalitat Valenciana, 2015). Potential direct economic damage costs are obtained using information on land use categories to define asset values and applying a depth-damage function, which estimates the expected damage for a given inundation depth.
  - Risk modeling.
- o Input data on floods (exceedance probabilities), river discharge (system response) and estimated consequences (aggregated outcomes at municipality scale from GIS-data) is incorporated into the risk model to estimate societal and economic risk in terms of annual expected impacts.

Table 1 shows a summary of most relevant variables and data sources for flood risk analysis based on the presented framework in Fig.1.

**2.7 Phase VII: Risk calculation**

Aggregated data on consequence estimation per flood event, from Phase VI, is incorporated into the risk model, proposed in Phase V. The iPresas UrbanSimp (www.ipresas.com) software tool is used for risk calculation and modelling. This tool is a simplified version of iPresas Calc, first developed by the Polytechnic University of Valencia (UPV) and now by iPresas Risk Analysis (Spin-off UPV). iPresas Calc is a software tool that combines input data on flood hazard and impact to obtain expected annual risk (Serrano-Lombillo et al., 2011). Risk, in terms of expected annual societal or economic risk, is calculated by developing the event tree that considers all combinations of events that may lead to flooding.

In this paper, flood risk is defined as the combination of the probability of a damaging flood event and potential consequences (Gouldby and Samuels, 2005; Schanze, 2006). Risk is estimated as the expected annual average damage of flooding in terms of societal or economic consequences. Hence, risk is obtained in terms of expected annual population affected (EAPA), number of injuries (EANI), fatalities (EAF) and damage costs (EAD). The iPresas UrbanSimp software tool estimates risk by developing the event tree that includes all combinations of flood events, system response and related consequences.

Although there are examples of flood risk analysis approaches which include economic, social and environmental risks (Meyer et al., 2009), conducting a quantitative analysis of environmental risks was out of scope of this research work. On the other hand, societal risk is considered based on a three-fold perspective: potential affected population, injured and fatalities.

**2.8 Phase VIII: Risk representation**

5  Risk can be represented in F-N curves. The area under the curve is the annual expected number of fatalities, where the horizontal axis represents the level of consequences (e.g. number of fatalities, denoted as N) and the vertical axis represents the annual cumulative probability of exceedance (F) of each level of consequences.

Other type of consequences can be represented. These curves are then called F-D or F-AP, by representing economic costs due to damages (D) or affected population (AP), respectively.

10  ## 2.9 Phase IX: Risk evaluation

Risk outcomes can be compared with tolerability recommendations (if available), thus enabling to analyze whether risk reduction measures are justified or not when evaluated in contrast with proposed criteria. Generalized frameworks for risk evaluation can be found in the literature (UK Health and Safety Executive, 2001). However, there still is a lack of tolerability criteria applied at local scale, although some recent examples can be found (Miller et al., 2015).

15  Tolerability recommendations for individual and societal risk have been published by several authors and organisations (Vrijling, 2001). As an example, the United States Bureau of Reclamation suggests a limit of 0.01 fatalities per year for annualized societal risk when analysing incremental risk from flooding due to dam failure (Hennig et al., 1997). However, this limit, proposed for analysing incremental risks (attributed to the failure of the infrastructure), may not be applied when analysing flood risks in total terms (due to all potential flood events from both failure and non-failure cases).

20  ## 2.10 Phase X: Study of risk reduction measures

Once risk is obtained for the Base Case, other scenarios can be analyzed to evaluate the impact of risk reduction measures. New input data on loads, system response or consequences should be required and incorporated into the risk model. Risk outcomes for the new scenario are then compared with results for the Base Case.

Regarding evaluation of risk reduction measures, approaches such as Cost-Benefit Analysis (CBA) and Multi-Criteria
25  Analysis (MCA) are commonly used for analyzing investment projects, required by law or promoted through guidelines (EC, 2008). In contrast to CBA, which is legally prescribed in some countries (for example in the Netherlands or the United Kingdom), MCA is not widely established, although some examples can be found. For example, project selection for public works in Italy and acquisition of data-processing equipment or consulting services by public administration in Spain have to be conducted based on MCA (Gamper and Turcanu, 2007).

In dam safety management, the use of risk indicators that consider efficiency and equity principles is common, evaluating societal and economic risk reduction and costs of measures. The Adjusted Cost per Statistical Life Saved (ACSLS) indicator is commonly used for evaluating dam risk reduction measures (Morales-Torres et al., 2016).

**3 Case study analysis**

5    An example of how the framework described in Sect.2 can be applied is included in this section. The results have been used to guide the development and implementation of a local action plan for flood risk management.

**3.1 Phase I: Scope of the case study**

The municipality of Oliva is located in the eastern coast of Spain (Fig. 4), has about 27,127 inhabitants (distributed in several urbanized areas) and covers a total area of 60.1km².

10    The heaviest daily precipitations historically observed in Spain concentrate mainly on the coastal Mediterranean zone. Indeed, Oliva accounts for the most extreme daily precipitation record in the Iberian Peninsula with 817 mm on 3$^{rd}$ November 1987 (Ramis et al., 2013).

The mean annual precipitation reaches 850 mm. Flood events concentrate mainly during the rainy season from August to November. Table 2 shows a summary of most relevant flood events in Oliva.

15    The system is characterized by multiple river courses and brooks, with complex interconnections and a varying topography, including low-land areas and hills up to 460 m.a.s.l.

A dam is currently under construction in Rambla Gallinera river course (a 62.5 m high concrete gravity dam, with a total reservoir capacity of 6.13 hm³ at dam crest level). Civil works started in 2010 (including river embankments, diversion of secondary brooks to Rambla Gallinera river course and dam construction), but are not finished yet. The dam will provide

20    flood protection up to a return period of 10 yr (Hijós Bitrián et al., 2010) and significant reduction on the peak flow discharges at this river course up to 56% (50-yr flood event). Discharges are also attenuated for floods with higher return periods, with a minimum reduction of 8.6% (5,000-yr flood).

Oliva is composed by several urbanized areas distributed within the municipality. The main area is located in the north-western part, concentrating 84.6% of resident population (59.6% of seasonal population). However, other areas located along

25    the coast are relevant as population may increase by 23 times in some districts.

The selection of this study area is based on several reasons. First, the intensity and frequency of past flood events in the region are relevant. Second, good quality and up-to-date data are available on hazard, population and land use mapping. Additionally, the impact of structural and non-structural flood risk reduction measures has not been quantified so far. Finally, local authorities are currently involved in the process of developing the Municipal Action Plan against Flood Risk (denoted

30    as PAMRI).

**3.2 Phase II: Review of available data**

Population and land use data are GIS-based. These data, provided by local government, is based on a yearly survey promoted by the regional government for all municipalities with less than 50,000 inhabitants (hereafter, EIEL database, by its acronym in Spanish). The municipality is distributed in 9,324 and 16,131 parcels of urban and rural land, respectively.

5 The EIEL database includes resident and seasonal population: "resident population" is obtained from census data and "seasonal population" is estimated from demographic trends observed in the last years during the summer season. It includes both resident and occasional population (but does not include hotel and campsite occupancy). For this analysis, the summer period ranges from mid-April to mid-September.

This database is completed with observations during site visits and other inputs from local authorities.

10 ### 3.3 Phase III: Study of the system situation: definition of the Base Case

Four scenarios are considered for flood risk analysis as follows:

— Current situation (Scenario 0): this scenario represents the current situation of the system and it is used for benchmarking (to compare with results of Scenarios 1 to 3). This scenario is considered as the Base Case.

— Implementation of structural measures (Scenario 1): this scenario represents the situation after implementing
15 structural measures for flood risk reduction, including dam construction. Differences in peak flow discharges in Rambla Gallinera are shown in Table 3 (e.g. from 282 to 182 m³/s for a 25-yr flood event).

— Implementation of a local action plan (Scenario 2): this scenario represents the situation after implementing a local action plan against flooding (PAMRI), which includes improved warning and communication schemes, public education campaigns and training of all actors involved in emergency management.

20 — Implementation of both local action plan and structural measures (Scenario 3): this scenario represents the situation after implementing both structural and non-structural measures.

**3.4 Phase IV: Flood events to be analysed**

[revised manuscript text omitted]

Estimation of potential life-loss is based on the method proposed by DEFRA (DEFRA, 2006). The number of fatalities is a function of the number of injuries and the hazard rating, where the number of injuries is estimated by combining the following factors:

- Number of people within the hazard zone.
- Hazard rating: factor which combines flood characteristics (flood depth, flow velocity and debris factor).
- Area vulnerability: function of effectiveness of flood warning, speed of onset of flooding and nature of area (including types of buildings); and
- People vulnerability: function of presence of people who are very old and/or infirm/disabled/long-term sick.

The following assumptions for the case study analysis are considered:

- An average debris factor (DF) equal to 0.5 is used to estimate hazard rates.
- A vulnerability area factor (AV) equal to 6, 7 and 8 is used for multi-storey buildings, residential areas and campsites, respectively.
- A population vulnerability factor (Y) of 0.2 is used based on census data (i.e. percentage of population aged 65 years and over).

We calculated the potential direct economic damage using information on land use classes (a summary is included in Table 5), reference asset values and a generic depth-damage function (denoted as CS in Fig. 8), which estimates the expected damage for a given inundation depth.

It is essential to adjust asset values to the regional economic situation and property characteristics (Jongman et al., 2012). Therefore, asset values and a generic stage-damage function used in regional studies for flood risk planning are considered in this case study (Generalitat Valenciana, 2015). A sensitivity analysis has been included to analyze their impact on results. Different stage damage functions would impact on consequence estimation results as later described in Sect.4.

Other direct costs such as destruction of vehicles, damage to infrastructure, livestock or business interruption are not considered. Indirect costs are considered based on factors used by regional planning (Generalitat Valenciana, 2015), set as 7% of direct costs for the city of Oliva (it includes aspects such as population, employment and number of households within the urban area).

Table 6 summarizes the results of consequence estimation. A 500-year flood could cause roughly 9 potential fatalities and 52M€ for Scenario 0.

The impact of implementing a local action plan against flooding (PAMRI) is analyzed based on the following changes on consequence estimation from improved warning systems and communication schemes:

- A lower rate of vulnerability area factor (AV) is considered. Hence, values change to AV=5 in urbanized areas with multi-storey buildings, AV=6 in residential areas and AV=7 in campsites.

‒ A reduction on economic damages is assumed based on damage avoided when a warning lead time of, at least, 2 hours is provided. For a 80% rate of warning coverage (proportion of covered properties), 100% rate of service effectiveness (proportion of flooded serviced properties that were sent a timely, accurate and reliable flood warning), 80% rate of availability (proportion of flooded services properties that received warning), 85% rate for ability (proportion of residents able to understand and respond to such a warning), and 85% rate for effective action (proportion willing to take effective action or which have actually taken effective action), a percentage of damage reduction of 18% is assumed for flood depths below 1.2 m (Parker et al., 2005).

**3.7 Phase VII: Risk calculation**

The iPresas UrbanSimp software tool is used to estimate risk by developing the event tree that includes all combinations of flood events, system response and related consequences.

Table 6 shows results in terms of expected annual population affected (AEAP), number of injuries (AENI), fatalities (AEF) and damage (AED). Risk outcomes for the current situation show societal risk levels up to 2,370 of annual expected affected population and 0.56 fatalities per year. Considerable risk reduction can be achieved by implementing planned structural measures (Scenario 1) thus societal risk would be reduced to 1,168 inhabitants per year (AEAP) and 0.28 fatalities per year (AEF). Affected population remains equal after implementing local action planning (Scenario 2) but societal risk in terms of potential fatalities would be reduced to 0.48 fatalities per year.

In addition, results reflect the combined effect of both structural and non-structural measures (Scenario 3). Societal risk after dam construction and implementation of the local action plan might change from 0.56 to 0.24 fatalities per year. Economic risk in terms of annual expected damages would vary from 6.11 to 1.89 M€ per year.

It is noted that at this stage, only direct benefits (such as the reduction in flood damage and improved warning systems) are included in the analysis of the impact of implementing a local action plan. Other benefits such as improved risk awareness or reduction on economic damages to vehicles and local businesses could be considered in future analyses.

**3.8 Phase VIII: Risk representation**

Figure 7 shows F-AP, F-N and F-D curves for all scenarios. The first graph depicts the cumulative annual exceedance probability (F) of each level of potential affected population (AP). Results show that there is a probability of $10^{-2}$ of exceeding 8,300 affected people due to flooding for the scenario with structural measures. This value is higher when considering the current situation, with approx. 11,300 affected people for the same probability. The second graph depicts the cumulative annual exceedance probability (F) of each level of potential fatalities (N). Results show that there is a probability of $10^{-2}$ of exceeding 3 fatalities for the current situation (Scenario 0). This value decreases after implementing structural measures (Scenario 1) to approx. 2 and up to 1.6 for combined structural and non-structural measures (Scenario 3). The third graph shows potential economic damages (D) with a probability of $10^{-2}$ of exceeding 28 M€ for the current situation

(Scenario 0). This value might decrease up to approx. 17 M€ after implementing combined structural and non-structural measures (Scenario 3).

Finally, results from risk analysis were represented in different hazard and risk maps to support local action planning against flood risk. Recommendations published by the RISKMAP project (www.risk-map.org) have been considered for elaborating these maps. An example is provided as supplementary material to this paper (affected population for the 500-yr flood event for the current situation).

**3.9 Phase IX: Risk evaluation**

Tolerability recommendations are not considered for this case study since there are no proposed criteria or guidelines at regional or national level in Spain.

**3.10 Phase X: Study of risk reduction measures**

The ACSLS indicator is obtained to evaluate cost-efficiency of analyzed measures. Table 7 shows implementation, maintenance and annualized costs for considered measures (local action plan and structural measures including dam construction). Results show that any of these measures would be justified in terms of efficiency on risk reduction since results show negative values (reduction of economic risk is higher than annualized costs). After implementing the local action plan (lowest ACSLS value), the resulting ACSLS indicator still remains negative when risks before and after implementing structural measures are compared, thus supporting the decision of also implementing planned structural measures.

[revised manuscript text omitted]

**5. Discussion**

The proposed framework and its application to a real case study in Spain shows how risk analyses provide information to gain knowledge about the system, the potential flood events that can happen and their consequences. Hence, risk analyses, as presented in this article, inform decision-makers, however may not capture all aspects of risk and uncertainties that may be important for making effective decisions. Therefore, in this section, limitations of the proposed framework and implications of flood risk analysis outcomes to local action planning are described, along with recommendations for improved flood risk analysis.

**5.1 Limitations**

The analysis framework used in this study is relatively straightforward, but it does allow to analyze risk and to assess the impact of different scenarios. It is proposed as a framework for enhancing local flood risk analysis at regional and national scale, potentially transferable to other local applications in Europe.

However, the following remarks are made:

— Type of flooding. In this paper, we analyzed river flooding but integrating multiple hazards would be of high interest in future upgrades (e.g. to analyze the influence of sea water levels in boundary conditions).

— Flood hazard. It is recognized that over-estimations of annual risk between 33% and 100% have been reported in other studies when only three return periods are used (Ward et al., 2011). Therefore, results suggest that results for the case study could benefit from paying more attention to the potential damage caused by high-probability flood events. As shown in Sect. 4.1, high-probability flood event analysis would help to better adjust existing protection levels and would be of interest for future upgrades (e.g. return periods of 5-yr, 10-yr).

— Economic consequence estimation. A generic relative stage damage function is used for the case study, based on methods used for regional planning. In addition, due to the lack of statistical information on building-specific asset values, available rates by land use type have been used in this analysis, although more suitable for macro-scale flood damage evaluations. As shown in Sect.4.4, information on building typology at micro-scale would be of interest for future upgrades.

－ Life-loss estimation. Sources of uncertainty include lack of data on detailed building typology (to better estimate area vulnerability), human behaviour, effectiveness of warning systems, among other factors. Sensitivity analyses indicate that societal risk for this case study is dominated by population concentrated in high vulnerable areas and seasonal variability. As shown in Sect. 4.2, societal risk may range from 1,940 affected population/yr (low occupancy) to 2,529 affected population/yr (high occupancy), then requiring good knowledge of population variations during the year.

**5.2 Recommendations for flood risk analysis**

Based on results from this analysis, we recommend that quantitative risk analyses become the basis for developing local flood risk management plans. Specific recommendations include:

－ Upgrading hydraulic modeling to a broad set of flood events for hazard mapping, and analyzing not only river flooding but also pluvial or coastal flooding. As shown in Sect. 4.1, flood hazard mapping should be performed for high-probability flood events.

－ Improved data gathering on population characteristics and distribution at local scale. As shown in Sect. 4.2, detailed information on population distribution and variability is required to better analyse risk, including daily and seasonal variations.

－ Improved land use data gathering at local scale for better analyze life-loss and economic consequences from flooding. As shown in Sect. 4.4, obtaining detailed data on building and asset characteristics would enable economic consequence analysis at micro-scale then improving the definition of asset values and better estimating economic risks.

－ Defining standardized relative stage damage functions and reference costs at national scale. As shown in Sect. 4.4, both local and river basin flood risk analysis should consider the same method for economic consequence estimation to allow comparative analysis, to upgrade current and future flood risk plans and to develop cost-benefit analysis for prioritizing flood risk reduction measures.

Authors acknowledge the fact that micro-scale quantitative flood risk analysis may require advanced know-how and expertise on risk analysis. However, the development and application of methods such the presented work in this paper will help local authorities to guide future analysis. In most cases, resources have already been allocated for conducting flood hazard analysis, then just requiring updating or further upgrading based on aforementioned recommendations and their combination with consequence estimation analysis.

**5.3 Local action planning implications**

Results from the case study demonstrate its applicability and usefulness to support decision making for local action planning. As described in Sect. 3.10, implementation costs are lower than benefits in terms of economic risk reduction (ACSLS values are negative).

The application of the proposed framework for quantifying local flood risk for the city of Oliva represents a novel analysis in Spain.

The following recommendations were made to local authorities for defining strategies for local action planning, derived from outcomes of conducted flood risk analysis:

— Definition of specific public education campaigns for resident and seasonal population, with emphasis in high vulnerable groups (e.g. the elderly, schools and campsites): As described in Sect.3.6, lower vulnerable rates are assumed for Scenarios 2 and 3. This assumption should be supported by better public education and warning schemes.

— Definition of a procedure to formally reporting flood events, damages and effect of communication and evacuations procedure: required for future updates of hazard and vulnerability analysis. This would enable to validate assumptions concerning the impact on flood consequences (lower area vulnerability and damages) of implementing the local action plan.

— Verification of established communication schemes between regional and local authorities, and with emergency and civil protection services: needed to ensure effectiveness of non-structural measures for flood risk reduction (reduced damages bases on available warning times).

— Identification of potential locations for assembly points and helicopter landing sites have been set based on population clusters, hazard maps, and available evacuation routes: developed risk maps (examples are included in supplementary material) were used to identify potential locations. These sites should be verified and reviewed in future updates.

— Data gathering on additional urban characteristics (e.g. building typology, daily variability of population in industrial and commercial areas, etc.): to upgrade risk analyses and provide improved outcomes for decision making. As shown in Sect. 4.2., societal risk is highly influenced by seasonal variability.

— Impact of future flood risk mitigation measures: As shown in Sect. 3.10, new risk reduction measures might be planned and evaluated in accordance with the ACSLS indicator (e.g. aiming at reducing annual expected affected population). The proposed framework for flood risk analysis will allow updating in future reviews of the local action plan.

**6. Conclusions and the way forward**

Quantification of societal and economic flood risk is not required by current legislation in Spain and is relatively novel in local flood risk management as a result of a lack of guidance, standardized methods or tools for local flood risk analysis. Examples can be found in other countries such as in England and Wales (Hall et al., 2003), but are still scarcely applied in

5   Spain.

[revised manuscript text omitted]

EC: Guide to Cost Benefit Analysis of Investment Projects., 2008.

Escuder-Bueno, I., Castillo-Rodriguez, J. T., Zechner, S., Jöbstl, C., Perales-Momparler, S. and Petaccia, G.: A quantitative flood risk analysis methodology for urban areas with integration of social research data, Nat. Hazards Earth Syst. Sci., 12(9), 2843–2863, doi:10.5194/nhess-12-2843-2012, 2012.

European Parliament: DIRECTIVE 2007/60/EC OF THE EUROPEAN PARLIAMENT AND OF THE COUNCIL of 23 October 2007 on the assessment and management of flood risks, , (L 228), 27–34, 2007.

Gamper, C. D. and Turcanu, C.: On the governmental use of multi-criteria analysis, Ecol. Econ., 2, 298–307, doi:10.1016/j.ecolecon.2007.01.010, 2007.

Generalitat Valenciana: Plan de Acción Territorial sobre prevención del Riesgo de Inundación en la Comunitat Valenciana (PATRICOVA), Valencia., 2015.

Hall, J. W., Meadowcroft, I. C., Sayers, P. B. and Bramley, M. E.: Integrated Flood Risk Management in England and Wales, , (August), 2003.

5   Hennig, C., Dise, K. and Muller, B.: Achieiving Public Protection with Dam Safety Risk Assessment Practices, 1997.

Hijós Bitrián, F., Mañueco Pfeiffer, M. G. and Segura Notario, N.: Comité nacional español de grandes presas, in Congreso Nacional de Presas, edited by C. N. E. D. G. PRESAS., 2010.

Jongman, B., Kreibich, H., Apel, H., Barredo, J. I., Bates, P. D., Feyen, L., Gericke, A. and Neal, J.: Comparative flood damage model assessment : towards a European approach, Nat. Hazards Earth Syst. Sci., 12, 3733–3752, doi:10.5194/nhess-10   12-3733-2012, 2012.

Jonkman, S. N., Vrijling, J. K. and Vrouwenvelder, A. C. W. M.: Methods for the estimation of loss of life due to floods : a literature review and a proposal for a new method, Nat. Hazards, 46, 353–389, doi:10.1007/s11069-008-9227-5, 2008.

Klijn, F., Kreibich, H., Moel, H. De and Penning-rowsell, E.: Adaptive flood risk management planning based on a comprehensive flood risk conceptualisation, Mitig Adapt Strateg Glob Chang., (20), 845–864, doi:10.1007/s11027-015-15   9638-z, 2015.

MAGRAMA: Propuesta de mínimos para la metodología de realización de los mapas de riesgo de inundación, Madrid., 2013.

Marcotullio, P. J. and McGranahan, G.: Scaling Urban Environmental Challenges: From local to global and back, Earthscan with UNU-IAS and IIED., 2006.

20   Merz, B. and Thieken, Æ. A. H.: Flood risk curves and uncertainty bounds, , 437–458, doi:10.1007/s11069-009-9452-6, 2009.

Merz, B., Kreibich, H., Schwarze, R. and Thieken, A.: Review article " Assessment of economic flood damage ," , 1697–1724, doi:10.5194/nhess-10-1697-2010, 2010.

Meyer, V., Scheuer, S. and Haase, D.: A multicriteria approach for flood risk mapping exemplified at the Mulde river, 25   Germany, Nat. Hazards, 48, 17–39, doi:10.1007/s11069-008-9244-4, 2009.

Miller, A., Jonkman, S. N. and Ledden, M. Van: Risk to life due to flooding in post-Katrina New Orleans, Nat. Hazards Earth Syst. Sci., 15, 59–73, doi:10.5194/nhess-15-59-2015, 2015.

Morales-Torres, A., Serrano-Lombillo, A., Escuder-Bueno, I. and Altarejos-García, L.: The suitability of risk reduction indicators to inform dam safety management, Struct. Infrastruct. Eng., 2479(February), 30   doi:10.1080/15732479.2015.1136830, 2016.

Nakicenovic, N., Lempert, R. and Janetos, A.: Special Issue of Climatic Change on the framework for the development of new socioeconomics scenarios for climate change research, Clim. Change, doi:10.1007/s10584-013-0982-2, 2013.

Parker, D. J., Tunstall, S. and Wilson, T.: Socio-Economic Benefits of Flood Forecasting and Warning, in International conference on innovation advances and implementation of flood forecasting technology, Tromso (Norway)., 2005.

Penning-Rowsell, E. C., Priest, S. J., Parker, D. J., Morris, J., Tunstall, S. M., Viavatenne, C., Chatterton, J. and D., O.: Flood and Coastal Erosion Risk Management. A manual for economic appraisal, London., 2013.

Ramis, C., Homar, V., Amengual, A., Romero, R. and Alonso, S.: Daily precipitation records over mainland Spain and the Balearic Islands, Nat. Hazards Earth Syst. Sci., 13, 2483–2491, doi:10.5194/nhess-13-2483-2013, 2013.

5  Serrano-Lombillo, A., Escuder-Bueno, I., Membrillera-Ortuño, M. G. De and Altarejos-García, L.: Methodology for the Calculation of Annualized Incremental Risks in Systems of Dams, Risk Anal., 31(6), 1000–1015, doi:10.1111/j.1539-6924.2010.01547.x, 2011.

SPANCOLD: Technical Guide on Risk Analysis applied to management of Dam Safety, edited by Spanish National Commitee on Large Dams, Spanish Professional Association of Civil Engineers, Madrid (Spain)., 2012.

10  UK Health and Safety Executive: Reducing Risks: Protecting People - HSE's decision making process, Norwich., 2001.

UNISDR: Disaster Statistics in Europe - "Floods , droughts and storms: a major threat for European countries," Brussels., 2009.

Université Catholique de Louvain: EM-DAT Database: The OFDA/CRED International Disaster Database, [online] Available from: http://www.emdat.be/ (Accessed 1 October 2015), 2015.

15  USACE: Economic Guidance Memorandum (EGM) 01-03: Generic Depth-Damage Relationships., 2000.

Velasco, M., Cabello, À. and Russo, B.: Flood damage assessment in urban areas . Application to the Raval district of Barcelona using synthetic depth damage curves, , 9006(November), doi:10.1080/1573062X.2014.994005, 2015.

Vrijling, J. K.: Probabilistic design of water defense systems in The Netherlands, , 74, 337–344, 2001.

Ward, P. J., De Moel, H. and Aerts, J. C. J. H.: How are flood risk estimates affected by the choice of return-periods?, Nat.
20  Hazards Earth Syst. Sci., 11(12), 3181–3195, doi:10.5194/nhess-11-3181-2011, 2011.

Ward, P. J., Jongman, B., Weiland, F. S., Bouwman, A., Beek, R. Van, Bierkens, M. F. P. and Ligtvoet, W.: Assessing flood risk at the global scale: model setup , results , and sensitivity, Environ. Res. Lett., 044019, doi:10.1088/1748-9326/8/4/044019, 2013a.

Ward, P. J., Jongman, B., Weiland, F. S., Bouwman, A., van Beek, R., Bierkens, M. F. P., Ligtvoet, W. and Winsemius, H.
25  C.: Assessing flood risk at the global scale: model setup, results, and sensitivity, Environ. Res. Lett., 8(4), 044019, doi:10.1088/1748-9326/8/4/044019, 2013b.

Winsemius, H. C., Van Beek, L. P. H., Jongman, B., Ward, P. J. and Bouwman, A.: A framework for global river flood risk assessments, Hydrol. Earth Syst. Sci., 17(5), 1871–1892, doi:10.5194/hess-17-1871-2013, 2013.

| Risk component | Main variables | Data source | Risk outcome |
|---|---|---|---|
| Flood probability | Annual exceedance probability (AEP) | Hydrologic studies | |
| | Failure probabilities (system response) | System reliability analysis | Annual Expected Affected population (AEAP)
 Annual Expected Number of injured people (AENI)
 Annual Expected Number of potential fatalities (AEN)
 Annual Expected Economic damages (AED) |
| Exposure | Flood depth (y)
 Flow velocity (v)
 Flooded areas (AF)
 Debris factor (DF)
 Hazard ratings (HR) | Hydraulic modelling | |
| Vulnerability | Area vulnerability (AV)
 People vulnerability (Y)
 Affected population (AP)
 Percent of damages (PD)
 Reference costs (CR)
 Number of injured people (NI)
 Number of potential fatalities (N)
 Economic damages (D) | Land use distribution
 Census data distribution
 Consequence analysis | |

**Table 1. Summary table of main variables and outcomes used in the presented framework for flood risk analysis.**

| Date (yyyy-mm-dd) | 1987-11-03 | 1997-12-04 | 1972-11-29 | 1997-06-18 | 2002-03-30 | 1996-09-11 | 2002-05-05 |
|---|---|---|---|---|---|---|---|
| Precipitation in 24 h (mm) | 817 | 378 | 354 | 288 | 220 | 197 | 188 |

**Table 2. Summary of recent (most relevant) flood events for the case study.**

| River course | Current situation (Scenario 0) | | | Structural measures (Scenario 1) | | |
|---|---|---|---|---|---|---|
| | Return period (yr) | | | | | |
| | 25 | 100 | 500 | 25 | 100 | 500 |
| Piles | 84 | 153 | 247 | 84 | 153 | 247 |
| Fonts | 54 | 107 | 186 | 54 | 107 | 186 |
| Algepsar | 7 | 11 | 23 | 7 | 11 | 23 |
| Frares | 4 | 7 | 16 | 4 | 7 | 16 |
| Alfadalí | 21 | 34 | 82 | 21 | 34 | 82 |
| Cementeri | 2 | 4 | 8 | 2 | 4 | 8 |
| Gallinera | 282 | 462 | 1025 | 182 | 284 | 829 |
| Benirrama | 16 | 28 | 63 | 16 | 28 | 63 |
| Bullent | 102 | 173 | 399 | 102 | 173 | 399 |
| Molinell | 84 | 146 | 318 | 84 | 146 | 318 |

**Table 3. Simulated peak flow discharges per river course (SOBEK model) [m³/s].**

| Land use type | Area (m²) | Percentage of urban area (%) | Reference value (EUR/ m²) |
|---|---|---|---|
| Commercial | 19,348 | 0.4% | 34.55 |
| Cultural | 47,916 | 1.0% | 34.55 |
| Health services | 23,672 | 0.5% | 34.55 |
| Industrial | 687,372 | 14.9% | 11.25 |
| Institutional | 17,288 | 0.4% | 34.55 |
| Office building | 14,573 | 0.3% | 34.55 |
| Other uses | 131,249 | 2.8% | 0 |
| Residential | 3,043,656 | 66.0% | 68.7 |
| Restaurants | 8,512 | 0.2% | 34.55 |
| Sports facilities | 614,618 | 13.3% | 34.55 |
| Warehouse | 58,112 | 1.3% | 11.25 |

**Table 4. Summary of land use types for the case study (non-urbanized areas were also analysed but not included in this table).**

| Season | Return period (yr) | Current situation (Scenario 0) | | | | Structural measures (Scenario 1) | | | |
|---|---|---|---|---|---|---|---|---|---|
| | | AP | NI | N | D (M€) | AP | NI | N | D (M€) |
| Summer (seasonal population) | 25 | 7795 | 85 | 2 | 10.86 | 5596 | 59 | 1 | 5.27 |
| | 100 | 13269 | 158 | 3 | 22.20 | 9850 | 109 | 2 | 12.15 |
| | 500 | 22890 | 341 | 9 | 52.03 | 18754 | 270 | 7 | 42.39 |
| Winter (resident population) | 25 | 1873 | 25 | 1 | 10.86 | 1572 | 22 | 0 | 5.27 |
| | 100 | 3428 | 51 | 1 | 22.20 | 2539 | 35 | 1 | 12.15 |
| | 500 | 6282 | 110 | 3 | 52.03 | 4497 | 80 | 2 | 42.39 |
| | | Local action plan (Scenario 2) | | | | Structural measures and local action plan (Scenario 3) | | | |
| | | AP | NI | N | D (M€) | AP | NI | N | D (M€) |
| Summer (seasonal population) | 25 | 7795 | 73 | 1 | 9.91 | 5596 | 51 | 1 | 4.73 |
| | 100 | 13269 | 136 | 3 | 20.26 | 9850 | 94 | 2 | 10.95 |
| | 500 | 22890 | 293 | 8 | 47.61 | 18754 | 232 | 6 | 38.92 |
| Winter (resident population) | 25 | 1873 | 22 | 0 | 9.91 | 1572 | 19 | 0 | 4.73 |
| | 100 | 3428 | 44 | 1 | 20.26 | 2539 | 30 | 1 | 10.95 |
| | 500 | 6282 | 95 | 3 | 47.61 | 4497 | 69 | 2 | 38.92 |

Note: AP=Affected population; NI=number of injured people; N=fatalities; D=damage costs in M EUR.

**Table 5. Estimated impact per scenario and flood event.**

| | Current situation (Scenario 0) | Structural measures (Scenario 1) | Local action plan (Scenario 2) | Structural measures and local action plan (Scenario 3) |
|---|---|---|---|---|
| Societal risk (AEAP) [inhabitants/yr] | 2370 | 1168 | 2370 | 1168 |
| Societal risk (AENI) [injured inh./yr] | 28 | 21 | 24 | 18 |
| Societal risk (AEN) [fatalities/yr] | 0.56 | 0.28 | 0.48 | 0.24 |
| Economic risk (AED) [Million EUR/yr] | 6.11 | 2.10 | 5.57 | 1.89 |

Note: AE=annual expected; AP=Affected population; NI=number of injured people; N=fatalities; D=damage costs.

**Table 6. Results from risk model per scenarios CS, BC and effect of local action plan.**

| Measure | Structural measures (Scenario 1) | Local Action Plan (Scenario 2) | Local Action Plan + Structural measures (Scenario 3) |
|---|---|---|---|
| Discount rate (%) | 5 | 5 | 5 |
| Life span (years) | 50 | 5 | 50 |
| Implementation cost (EUR) | 43,000,000 | 10,000 | 43,000,000 |
| Maintenance cost (EUR/yr) | 10,000 | 2,500 | 10,000 |
| Annualized cost (EUR/yr) | 2,253,238 | 4,700 | 2,253,238 |
| Annualized cost | 2.25 | 0.00 | 2.25 |
| ACSLS (MEUR/life) [compared with current situation] | -6.27 | -6.69 | NA |
| ACSLS (MEUR/life) [compared with situation after implementing Local Action Plan] | NA | NA | -5.94 |

Note: ACSLS=Adjusted Cost per Statistical Life Saved, NA=Not Applicable.

**Table 7. Results from evaluation of risk reduction measures.**

| | Oliva (Scenario 0: CS) | | Valencia (region) | | Comparison Local/Region | |
|---|---|---|---|---|---|---|
| Flood protection level (yr) | Societal risk (AEAP) [inhabitants/yr] | Economic risk (AED) [MEUR/yr] | Societal risk (AEAP) [inhabitants/yr] | Economic risk (AED) [MEUR/yr] | %AEAP | %AED |
| 1 | 2370 | 6.11 | No data | No data | - | - |
| 2 | 2279 | 5.88 | 47600 | 746.24 | 4.8% | 0.8% |
| 5 | 1991 | 5.16 | 29000 | 537.94 | 6.7% | 0.9% |
| 10 | 1557 | 4.07 | 15800 | 348.04 | 9.8% | 1.2% |

Note: CS=current situation; AEAP=annual expected affected population; AED=annual expected damage costs.
**Table  8. Effect of the selection of flood protection level.**

| Land use type | Warehouse | Commercial | Cultural | Industrial | Office | Households | Health services | Agricultural |
|---|---|---|---|---|---|---|---|---|
| GVA | 11.25 | 34.55 | 34.55 | 11.25 | 34.55 | 68.7 | 34.55 | 0.8 |
| MAGRAMA | 150 | 380 | 200 | 450 | 380 | 350 | 200 | 5 |

**Table 9. Reference costs in EUR/m² in urban areas: GVA (2015) and MAGRAMA (2013).**

[Figure]

**Figure 1: Flowchart of data and models.**

[Figure]

**Figure 2: Generic division of the analyzed range of flood events.**

[Figure]

(a)

(b)

**Figure 3: Generic risk model architecture: non-regulated river system (a) and regulated river system (b).**

[Figure]

**Figure 4: Location of the case study area at national (left) and regional (right) scale.**

[Figure]

**Figure 5: Risk model architecture for the case study of Oliva using iPresas UrbanSimp software tool.**

[Figure]

**Figure 6: Hazard level map for Scenario 0 (Current situation).**

[Figure]

**Figure 7: Extract of F-AP, F-N and F-D curves for the case study: scenarios 0 (current situation), scenario 1 (PAMRI=local action plan), scenario 2 (structural measures), and scenario 3 (PAMRI plus structural measures).**

[Figure]

**Figure 8: Examples of depth-damage functions compared to function used for the case study analysis (CS).**